# Displacement-Resistant Extensions of DPO with Nonconvex $f$-Divergences

**Idan Pipano**
Faculty of Data and Decision Sciences
Technion - Israel Institute of Technology
idan.pipano@campus.technion.ac.il

**Shoham Sabach**
Faculty of Data and Decision Sciences
Technion - Israel Institute of Technology
ssabach@cornell.edu

**Kavosh Asadi**
Meta
kavosh@meta.com

**Mohammad Ghavamzadeh**
Qualcomm AI Research
ghavamza@qti.qualcomm.com

## Abstract

DPO and related algorithms align language models by directly optimizing the RLHF objective: find a policy that maximizes the Bradley-Terry reward while staying close to a reference policy through a KL divergence penalty. Previous work showed that this approach could be further generalized: the original problem remains tractable even if the KL divergence is replaced by a family of $f$-divergence with a convex generating function $f$. Our first contribution is to show that convexity of $f$ is not essential. Instead, we identify a more general condition, referred to as DPO-inducing, that precisely characterizes when the RLHF problem remains tractable. Our next contribution is to establish a second condition on $f$ that is necessary to prevent probability displacement, a known empirical phenomenon in which the probabilities of the winner and the loser responses approach zero. We refer to any $f$ that satisfies this condition as displacement-resistant. We finally focus on a specific DPO-inducing and displacement-resistant $f$, leading to our novel SQUAREDPO loss. Compared to DPO, this new loss offers stronger theoretical guarantees while performing competitively in practice.

## 1 Introduction

Language models (LMs) have emerged as a promising path towards achieving AGI. As they become increasingly integrated into real-world applications, it is important to ensure that LMs behave in accordance with human preferences. A major step towards enabling LM alignment was the reinforcement learning from human feedback (RLHF) framework (Christiano et al., 2017). In this context, the alignment problem is formulated as maximizing a learned Bradley-Terry (BT) reward signal while ensuring that the final LM does not deviate too much from a reference policy. This is a guardrail achieved by penalizing deviations through a KL-divergence penalty. A breakthrough was due to Rafailov et al. (2024) who showed that the RLHF problem could be solved directly in a single step by leveraging the fact that (I) the problem has a closed-form solution, and (II) plugging the closed-form solution into the BT model removes intractable quantities, leading to simple optimization.

More recent work showed that the so-called DPO approach of Rafailov et al. (2024) can be generalized. In particular, Wang et al. (2024) showed that it is possible to maintain tractable optimization while moving from the KL divergence to a family of $f$-divergences. This use of $f$-divergences, not to be confused with minimizing the $f$-divergence between the learned model and a desired model (Go et al., 2023; Han et al., 2024), is the focus of our work. More precisely, Wang et al. (2024) showed that if $f$ is convex, differentiable, and $f'$ is invertible with $0 \notin dom(f')$, then the new optimization problem remains directly solvable, akin to solving the original RLHF problem using DPO.

Our first contribution is to show that the RLHF problem remains tractable for even a broader class of functions than those discovered by Wang et al. (2024). In particular, we show that surprisingly one can relax the convexity assumption and that even for some non-convex functions $f$, one can get $f$-divergences that maintain tractability. In fact, we discover the full characterization of functions $f$ for which the original RLHF problem remains tractable. Referred to as DPO-inducing, the new condition is less restrictive than the class of functions discussed by prior work.

The richness of the class of DPO-inducing functions gives us significant flexibility, but also raises a natural question: which specific choice of DPO-inducing $f$ is best equipped to improve alignment? To answer this question, we note that an empirical pitfall of DPO and related algorithms is *probability displacement*, a phenomenon in which both winner and loser probabilities approach zero during optimization (Fisch et al., 2025). Our second contribution is to show that within the class of DPO-inducing functions, there exists a subset of functions that provably hedge against probability displacement. We refer to such functions as displacement-resistant.

We finally focus on a novel loss, called SQUAREDPO, which arises by using a nonconvex $f$-divergence that is both DPO-inducing and displacement-resistant. We show that optimizing this novel loss results in comparative empirical performance while enjoying robustness to over-optimization and empirically mitigating displacement. Together, our results suggest that DPO-inducing $f$-divergences can be used effectively for preference optimization, and that displacement-resistant $f$-divergences, in particular, are best suited for aligning LMs.

## 2 PRELIMINARIES

A key step in training Language Models (LMs) is to align them with human preferences. In this task, we leverage a *preference dataset* $\mathcal{D}$, often consisting of triplets $(x, y_w, y_l)$. Here, $x$ is a prompt, and $y_w, y_l$ are two responses, where a human, or a strong LLM, has annotated $y_w$ to be a better response for the prompt $x$ than $y_l$. The response $y_w$ is referred to as the *chosen* or *winner* response, and $y_l$ as the *rejected* or *loser* response.

The original technique to solve this task was to perform a two-stage optimization (Christiano et al., 2017). Namely, we first train a parameterized reward model $r_\phi$ by minimizing the negative-log-likelihood loss:

$$\min_{\phi} \mathbb{E}_{(x, y_w, y_l) \sim \mathcal{D}} \left[ -\log \sigma \left( r_\phi \left( x, y_w \right) - r_\phi \left( x, y_l \right) \right) \right] , \tag{1}$$

under the assumption that our preferences follow the Bradley-Terry (BT) (Bradley & Terry, 1952) model, i.e., that there exists a reward function $r$ such that the probability of $y_w$ being preferred over $y_l$ given a prompt $x$ is $\sigma \left( r \left( x, y_w \right) - r \left( x, y_l \right) \right)$, where $\sigma \left( t \right) = 1 / \left( 1 + e^{-t} \right)$ is the sigmoid function. Here and throughout, $\log(\cdot)$ denotes the natural logarithm (base $e$). The learned reward model $r_\phi$ is then used by the PPO algorithm to optimize the following objective:

$$\max_{\theta} \mathbb{E}_x \left[ \mathbb{E}_{y \sim \pi_\theta(\cdot | x)} \left[ r_\phi \left( x, y \right) \right] - \beta \mathrm{KL} \left[ \pi_\theta \left( \cdot \mid x \right) \| \pi_{\mathrm{ref}} \left( \cdot \mid x \right) \right] \right] . \tag{2}$$

Here, higher values of $\beta > 0$ provide more incentive for the final policy to remain close to $\pi_{\mathrm{ref}}$, where $\pi_{\mathrm{ref}}$, called the reference policy, is the checkpoint from which optimization is initialized (often the SFT checkpoint).

Noting that Problem (2) admits a closed-form solution, Rafailov et al. (2023) derived a one-stage algorithm, called DPO, as an alternative for the two-stage RLHF optimization described above. Note that the closed-form solution of (2) is given by $\pi_\theta \left( y \mid x \right) = \pi_{\mathrm{ref}} \left( y \mid x \right) \exp \left( r_\phi \left( x, y \right) / \beta \right) / Z \left( x \right)$, where the so-called partition function $Z \left( x \right)$ does not depend on $y$. This closed-form solution can be rewritten as:

$$r_\phi \left( x, y \right) = \beta \log \frac{\pi_\theta \left( y \mid x \right)}{\pi_{\mathrm{ref}} \left( y \mid x \right)} + \beta \log Z \left( x \right) . \tag{3}$$

Substituting this into (1) leads to the single-stage loss:

$$\min_\theta \left\{ \mathcal{L}_{\text{DPO}}\left(\theta\right) := \mathbb{E}_{(x,y_w,y_l)\sim\mathcal{D}} \left[ -\log \sigma \left( \beta \log \frac{\pi_\theta\left(y_w \mid x\right)}{\pi_{\text{ref}}\left(y_w \mid x\right)} - \beta \log \frac{\pi_\theta\left(y_l \mid x\right)}{\pi_{\text{ref}}\left(y_l \mid x\right)} \right) \right] \right\} , \quad (4)$$

which can be minimized directly.

While simple and effective, DPO exhibits a counterintuitive empirical phenomenon. This so-called probability displacement phenomenon refers to the tendency of DPO to take the probability of both the preferred and the dispreferred responses to zero (Pang et al., 2024; Xiao et al., 2024; Shen et al., 2024; Nvidia et al., 2024; Wu et al., 2025; Pal et al., 2024; Rafailov et al., 2024; Fisch et al., 2025; D'Oosterlinck et al., 2025; Asadi et al., 2025; Razin et al., 2025). While the research on this topic is active and ongoing, the theoretical understanding remains partial. Our goal is to advance this understanding by identifying the precise requirements that ensure both tractability and stability of DPO optimization.

## 3 DPO WITH NON-CONVEX $f$-DIVERGENCES

Note that the RLHF objective (2) employs the KL divergence to measure the discrepancy between the learned policy and the reference policy. However, this discrepancy could be measured using a more general notion of distance, such as with the family of $f$-divergences (Wang et al., 2024; Huang et al., 2025) defined next (Csiszár, 1972).

**Definition 1.** *Let* $f : \mathbb{R}_+ \to \mathbb{R}$ *be a function*[1] *such that* $f\left(1\right) = 0$. *For two probability distributions* $\mathbf{p}, \mathbf{q} \in \Delta_n$, *the* $f$-*divergence between* $\mathbf{p}$ *and* $\mathbf{q}$ *is defined as*

$$D_f\left(\mathbf{p} \parallel \mathbf{q}\right) = \sum_{i=1}^{n} q_i f\left(p_i/q_i\right).$$

KL-divergence is an instantiation of $f$-divergence with $f_{\text{KL}}\left(t\right) = t\log t$. This naturally allows choosing various $f$-s to obtain various alternatives instead of the KL divergence in the RLHF problem. Therefore, we now focus on the following more general RLHF problem:

$$\max_{\pi_\theta} \mathbb{E}_{x\sim\mathcal{D}} \left[ \mathbb{E}_{y\sim\pi_\theta(\cdot|x)}\left[r_\phi\left(x,y\right)\right] - \beta D_f\left[\pi_\theta\left(\cdot \mid x\right) \parallel \pi_{\text{ref}}\left(\cdot \mid x\right)\right] \right] . \quad (5)$$

However, recall that our desire is to keep the original RLHF problem tractable and easy to optimize, and so the choice of $f$ needs to be made with this consideration in mind. Interestingly, Wang et al. (2024) show that for any convex $f$ with invertible $f'$ that satisfies $0 \notin \text{dom}\left(f'\right)$, substituting the optimal solution of the generalized RLHF problem (5) into the original BT model (1) yields the following DPO-like loss, referred to as $f$-DPO:

$$\min_\theta \left\{ \mathcal{L}_{f\text{-DPO}}\left(\theta\right) := \mathbb{E}_{(x,y_w,y_l)\sim\mathcal{D}} \left[ -\log \sigma \left( \beta f'\left( \frac{\pi_\theta\left(y_w \mid x\right)}{\pi_{\text{ref}}\left(y_w \mid x\right)} \right) - \beta f'\left( \frac{\pi_\theta\left(y_l \mid x\right)}{\pi_{\text{ref}}\left(y_l \mid x\right)} \right) \right) \right] \right\} . \quad (6)$$

In the next section, we show that convexity is not necessary, meaning that there exist non-convex $f$ for which the problem remains tractable.

### 3.1 WHICH $f$-S *Can* BE USED?

We now provide a characterization that fully specifies the class of functions $f$ for which the RLHF optimization problem is tractable. To this end, we introduce the following definition:

**Definition 2.** *We say a function* $f : \mathbb{R}_+ \to \mathbb{R}$ *is* DPO-inducing *if substituting any optimal solution of* (5) *into the BT model* (1) *yields the* $f$-*DPO loss* (6).

Our aim is then to characterize which functions $f$ are DPO-inducing and which are not. The complete characterization is provided in Theorem 1 in the Appendix. As the Theorem 1 is rather technically involved, in Corollary 1 we add a slight assumption (existence of the limit $\lim_{t\to 0^+} f'\left(t\right)$) under which the characterization becomes simpler and straightforward to check.

---

[1]We note that traditionally $f$-divergences are defined for convex functions $f$. However, as we shall see in § 3.1, in the context of direct preference optimization with $f$-divergence, convexity is not necessary.

**Corollary 1.** *[Proof in Appendix E.2] Let $f : \mathbb{R}_+ \to \mathbb{R}$ be a continuous function in $\mathbb{R}_+$, which is continuously differentiable in $\mathbb{R}_{++}$[2]. Assume that $\lim_{t \to 0^+} f'(t)$ exists (possibly $\pm\infty$). Then, $f$ is DPO-inducing if and only if $\lim_{t \to 0^+} f'(t) = -\infty$.*

See Figure 1, where functions inside and outside the DPO-inducing region are examples of functions that do and do not satisfy the condition in Corollary 1, respectively. An interesting point about the proof of Theorem 1 is that it uses the equivalence of the DPO-inducing property to a property we call *interior-inducing* (see Definition 3 in the Appendix). In short, $f$ is interior-inducing if optimal solutions to (5) assign non-zero probability to all responses.

The importance of this result is two-fold. First, it reveals a broader family of functions $f$ that can be used in $f$-DPO, not just convex functions as discussed by Wang et al. (2024). Second, the result shows that failure to satisfy $\lim_{t \to 0^+} f'(t) = -\infty$ means that the optimization problem will become intractable.

## 3.2 Which $f$-s *Should* Be Used?

Having characterized the class of DPO-inducing functions, i.e., the functions $f$ that *can* be used in $f$-DPO, we now show that some functions in this class are theoretically less favorable. To this end, given a preference dataset $\mathcal{D}$ and a prompt $x$ from the dataset, we denote by $S_x$ the set of responses $y$ that appeared in the preference dataset as a response for $x$ (either as a winner or as a loser). Consider the following optimization problem

$$\max_{\pi_\theta} \mathbb{E}_{x \sim \mathcal{D}} \left[ \mathbb{E}_{y \sim \pi_\theta(\cdot|x)} [r_\phi(x, y)] - \beta \sum_{y \in S_x} \pi_{\text{ref}}(y \mid x) f\left( \frac{\pi_\theta(y \mid x)}{\pi_{\text{ref}}(y \mid x)} \right) \right], \quad (7)$$

which differs from the modified RLHF objective (5) only by the summation in the $f$-divergence term being only over $S_x$ instead of over the entire response space. Note that Problem (7) might admit optimal solutions that are very different than those of Problem (5) (see Appendix D.2).

We begin with Lemma 1, which is interesting in its own right and, as we will see in Lemma 2, also plays a key role in understanding the likelihood displacement phenomenon.

**Lemma 1** (Proof in Appendix E.3). *Let $f : \mathbb{R}_+ \to \mathbb{R}$ be a DPO-inducing function. Then, substituting any optimal solution to (7) into the BT model (1) yields the same $f$-DPO loss (6) as obtained from substituting any optimal solution to (5) into the BT model.*

Lemma 1 reveals a counterintuitive and concerning property of $f$-DPO: while it solves the two RLHF losses (5) and (1), it is also true that it solves the two losses (7) and (1). One way to see why this result is concerning is that Problem (7) is conceptually problematic, as its regularization term acts only over in-sample responses $y \in S_x$, while intuitively we want to regularize $\pi_\theta$ to remain close to $\pi_{\text{ref}}$ over the entire response space. To gain additional insight, in Appendix D.2 we find the set of optimal solutions to this problem in the special case where $f$ is convex. Another aspect, which holds regardless of the convexity of $f$, is that based on the value of $\arg\min_{t \in \mathbb{R}_+} f(t)$, the optimal solutions to (7) could have undesirable properties, as Lemma 2 shows.

**Lemma 2** (Proof in Appendix E.4). *Let $f$ be a DPO-inducing function with a unique global minimum at $c \in (0, 1]$. Suppose $r_\phi(x, y) \leq \max_{y' \notin S_x} r_\phi(x, y')$ for all $y \in S_x$. Then, any optimal solution to (7) satisfies*

$$\pi_\theta(y \mid x) \leq c \cdot \pi_{ref}(y \mid x), \quad (8)$$

*for any prompt $x$ and in-sample response $y \in S_x$.*

Lemma 2 shows that under mild assumptions[3], if we use any $f$ for which $\arg\min_{t \in \mathbb{R}_+} f(t) < 1$, optimal solutions to (7) suffer from likelihood displacement. As a special case, consider $f_{\text{KL}}(t) = t \log t$, which is the generating function for the KL divergence and hence corresponds to the original loss of DPO. Since $\arg\min_{t \in \mathbb{R}_+} f_{\text{KL}}(t) = e^{-1}$, we get a probability decrease in the in-sample

---

[2]We denote $\mathbb{R}_+ = [0, \infty)$ and $\mathbb{R}_{++} = (0, \infty)$.

[3]To see why the assumption in Lemma 2 is mild, notice that it is equivalent to requiring the existence of a response $y' \notin S_x$ with higher reward than $y$. As $S_x$ typically consists of only two responses, its complement covers almost the entire response space, making the condition very likely to hold.

responses by a factor of at least $e^{-1}$. Although a similar result was shown in previous work (Asadi et al., 2025), it was only for DPO, whereas here we prove this for general $f$. The importance of generalizing this statement to a general $f$ is that from Lemma 2 arises a theoretically grounded way to mitigate the likelihood displacement issue - choosing an $f$ for the $f$-divergence that satisfies $\arg\min_{t\in\mathbb{R}_+} f(t) \geq 1$. We refer to any $f$ that satisfies this property as a *displacement-resistant* function.

To summarize this section, our theory imposes two desiderata for the function $f$ of the $f$-divergence that nicely translate to easy-to-check mathematical properties: it should be DPO-inducing, i.e., $\lim_{t\to 0^+} f'(t) = -\infty$, and it should be displacement-resistant, i.e., $\arg\min_{t\in\mathbb{R}_+} f(t) \geq 1$. In the next section, we propose one specific $f$ that satisfies our two desiderata, discuss the resulting loss, and report empirical evaluations.

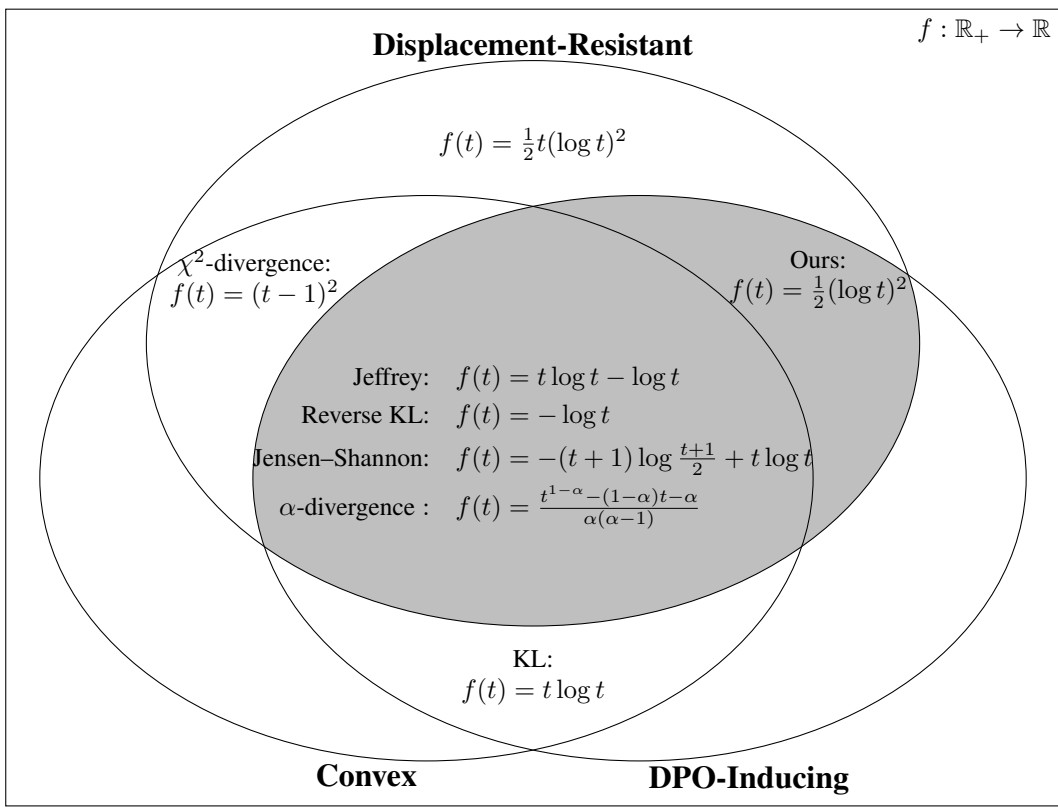

Figure 1: A Venn diagram illustrating a taxonomy of some generating functions $f$. The diagram includes classical examples of $f$-s used in $f$-divergences, as well as the functions $f(t) = \frac{1}{2}(\log t)^2$ and $f(t) = \frac{1}{2}t(\log t)^2$, which correspond to a Monte Carlo approximation of KL proposed by Schulman (2020). DPO-inducing refers to Definition 2 and displacement-resistant refers to the condition $1 \leq \arg\min_{t\in\mathbb{R}_+} f(t)$ proposed in §3.2 to mitigate likelihood displacement. The gray area is the intersection of these two sets of functions.

**A note on convexity**  Notice that as long as $f$ satisfies our two properties, there seems to be no reason to require $f$ to be convex. In fact, recall that a main reason to require $f$ to be convex in $f$-divergences is that together with Jensen's inequality and the assumption $f(1) = 0$, it guarantees that the divergence is non-negative. However, our result in Lemma 1 undermines this logic, as in Problem (7) the summation in the 'divergence' term is over a strict subset of the response space. As a result, Jensen's inequality does not apply, hence convexity of $f$ no longer guarantees the non-negativity of this term.

## 4  SQUAREDPO

Our proposed method is the instantiation of the $f$-DPO loss (6) obtained by choosing $f_{\text{SQUAREDPO}}(t) := (\log t)^2/2$. We refer to the resulting algorithm as SQUAREDPO. The function $f_{\text{SQUAREDPO}}$ is one possibility for a function that satisfies the two properties suggested in §3, as demonstrated in Figure 1. This particular function is both DPO-inducing and displacement-resistant thus making it a natural choice for preference optimization. Note that we deliberately chose a non-convex $f$, in order to explore the broader (compared to prior work) class of functions that Corollary 1 enables. We note that non-convex losses have proven effective in the past for preference alignment, e.g., LLM-proposed losses presented in Lu et al. (2024). However, we are the first to explore $f$-DPO (6) with non-convex functions $f$.

The explicit formula for the SQUAREDPO loss, obtained by substituting $f'_{\text{SQUAREDPO}}(t) = (\log t)/t$ in (6), is given by

$$\mathcal{L}_{\text{SquaredPO}} := \mathbb{E}_{(x,y_w,y_l)\sim\mathcal{D}}\left[-\log\sigma\left(\beta_\theta(y_w,x)\log\left(\frac{\pi_\theta(y_w|x)}{\pi_{\text{ref}}(y_w|x)}\right) - \beta_\theta(y_l,x)\log\left(\frac{\pi_\theta(y_l|x)}{\pi_{\text{ref}}(y_l|x)}\right)\right)\right], \tag{9}$$

where $\beta_\theta(y,x) := \beta/\frac{\pi_\theta(y|x)}{\pi_{\text{ref}}(y|x)}$, and $\beta > 0$ is as usual the regularization hyperparameter. Compared to the original DPO loss (4), the SQUAREDPO loss can be viewed as "DPO with adaptive $\beta$-s". There is a connection between our approach and some of the recent approaches (Meng et al., 2024; Wu et al., 2024; Lee et al., 2025) in that we both use adaptive $\beta$-s, but here the adaptive $\beta$-s are derived from theory and first principles rather than introduced heuristically. The adaptive $\beta$-s in SQUAREDPO can act as a safeguard against severe likelihood displacement; when $\pi_\theta(y_w\mid x)$ decreases, as is often observed in practice, $\beta_\theta(y_w,x)$ increases. An increase in the effective regularization coefficient intuitively means stronger regularization, thereby counteracting the decrease in probability.

## 5  EXPERIMENTS

In this section, we validate our theoretical findings regarding the ability of SQUAREDPO to mitigate likelihood displacement, demonstrate that it is robust to over-optimization compared to DPO, and that it performs competitively on standard benchmarks.

### 5.1  EXPERIMENTAL SETUP

**Model and Dataset**  Our reference model $\pi_{\text{ref}}$, i.e., the model from which we initialize preference optimization, is Meta-Llama-3-8B-Instruct (AI@Meta, 2024). The preference dataset we use to align $\pi_{\text{ref}}$ is TL;DR (Völske et al., 2017). We use a version of this dataset provided by von Werra et al. (2020) that is preprocessed for preference optimization. The dataset consists of Reddit posts from various topics, each paired with a preferred and a dispreferred summary.

**Evaluation**  We compare SQUAREDPO with DPO on three axes. First, we want to empirically probe our theoretical findings that SQUAREDPO is more resistant to likelihood displacement than DPO. This is done by computing the chosen log-ratios, $\log(\pi_\theta(y_w\mid x)/\pi_{\text{ref}}(y_w\mid x))$, for all training samples $(x,y_w,y_l)\in\mathcal{D}$, in different checkpoints throughout training. Second, we measure the performance of each method on the dataset's validation split, reporting win rates as judged by GPT-4. Lastly, to assess SQUAREDPO's ability to generalize to out-of-distribution tasks, we compare its performance against DPO on standard evaluation benchmarks.

The evaluation benchmarks we considered are AlpacaEval 2 (Li et al., 2023; Dubois et al., 2025) and MT-Bench (Zheng et al., 2023). AlpacaEval 2 contains $805$ single-turn prompts, where GPT-4 judges[4] compute both win rate (fraction of cases a model beats a strong baseline) and a length-controlled win rate that corrects for verbosity bias. MT-Bench consists of $80$ multi-turn questions across diverse categories, scored on a 1–10 scale by GPT-4.

---

[4]We use OpenAI's gpt-4o instead of the default judge gpt-4-1106-preview for reduced API costs. Dubois et al. (2025) show that the induced rankings are very robust to the choice of the judge.

## 5.2 Experimental Results

We align Meta-Llama-3-8B-Instruct on the TL;DR dataset over four training epochs using LoRA (Hu et al., 2022). Full hyperparameter and implementation details are provided in Appendix B.

**Performance on The Validation Set** We sample a set of $512$ examples from the validation split of the dataset. Inspired by the experimental setting in Huang et al. (2025), for each epoch in $\{1, 2, 4\}$ we use the checkpoint obtained at the end of this epoch to generate responses to each of the $512$ samples. We then take two approaches to assess the performance of our method. First, in Figure 2 we use GPT-4 to compute the win rate of SQUAREDPO against DPO. Second, in Table 1 we use GPT-4 to compute win rates of each of the two methods against the reference model, i.e. the model before alignment. In Table 1, we include another baseline, namely $\chi$PO (Huang et al., 2025).

In both approaches, it is evident that SQUAREDPO is robust to over-optimization. In Table 1 we see that while DPO's win rate against the reference model drops substantially below $50\%$ already in the second epoch, this is not the case for our loss. The gap in performance grows with the number of epochs. The table also shows that SQUAREDPO is more robust to over-optimization than $\chi$PO. In Figure 2 we see the same phenomenon from a different angle: when computing the win rate of SQUAREDPO against DPO, SQUAREDPO achieves statistically significant improvements over DPO in regimes prone to over-optimization (i.e., after two epochs or more).

One can see in Figure 2 and Table 1 that, albeit not in a statistically significant manner, DPO outperforms SQUAREDPO on the first epoch. In this context, it is worth emphasizing that the hyperparameters used for training were standard choices for DPO (e.g. $\beta = 0.01$), and we did not tune them for our loss. This suggests that with appropriate hyperparameter tuning, SQUAREDPO may potentially surpass DPO also in the first epoch.

Table 1: Win Rate on TL;DR's validation set against the base model Meta-Llama-3-8B-Instruct when trained on TL;DR. These results demonstrate SQUAREDPO's robustness to over-optimization compared to DPO and $\chi$PO.

| Epochs | SQUAREDPO winrate (%) | $\chi$PO winrate (%) | DPO winrate (%) |
|--------|------------------------|----------------------|------------------|
| 1 | $50.8 \pm 0.7$ | $51.2 \pm 0.8$ | $51.8 \pm 1.0$ |
| 2 | $50.6 \pm 1.1$ | $48.9 \pm 1.1$ | $45.0 \pm 1.1$ |
| 4 | $51.0 \pm 0.7$ | $48.3 \pm 1.1$ | $34.7 \pm 1.3$ |

**Standard Benchmarks** In Table 2 we present benchmark results for the models obtained by training Meta-Llama-3-8B-Instruct for one epoch on TL;DR with SQUAREDPO, $\chi$PO or DPO. Each AlpacaEval number is averaged over 10 seeds with confidence intervals. The MT-Bench score is an average over eight categories; the per-category breakdown is shown in Figure 5. Additional implementation details can be found in Appendix B. The results in Table 2 show that SQUAREDPO is almost on par with DPO on these benchmarks, with DPO slightly outperforming (it is worth emphasizing that we did not tune hyperparameters for SQUAREDPO). SQUAREDPO is on par with $\chi$PO on AlpacaEval and is more performant than $\chi$PO on MT-Bench.

**Likelihood Displacement Mitigation** To gauge SQUAREDPO's ability to mitigate likelihood displacement, i.e. the phenomenon where winner probabilities decrease during preference optimiza-

Table 2: Benchmark results for our SQUAREDPO and two baselines. For AlpacaEval 2, we report both raw win rate (WR) and length-controlled win rate (LC).

| Method | AlpacaEval 2 | | MT-Bench |
|--------|--------------|--------------|----------|
| | LC (%) | WR (%) | Score (1-10) |
| SQUAREDPO | $29.2 \pm 0.4$ | $24.5 \pm 0.3$ | 7.924 |
| $\chi$PO | $29.3 \pm 0.6$ | $24.3 \pm 0.4$ | 7.900 |
| DPO | $29.6 \pm 0.4$ | $24.8 \pm 0.4$ | 7.925 |

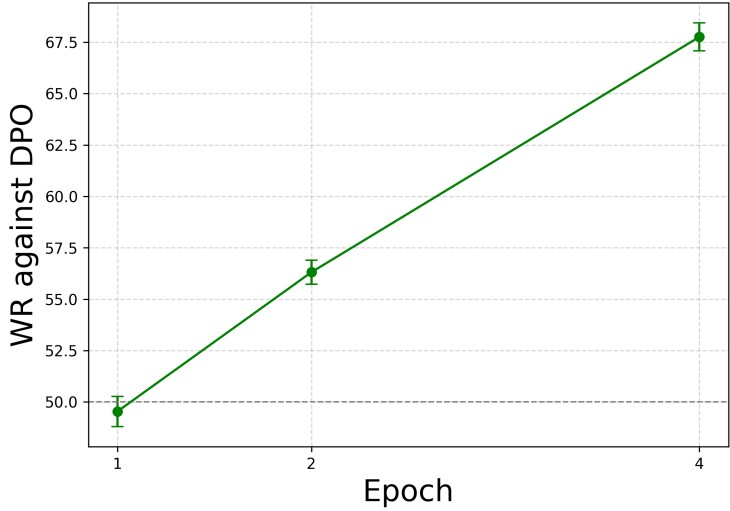

Figure 2: Head-to-head win rate of SQUAREDPO against DPO on TL;DR's validation split across training epochs when using these methods to finetune Meta-Llama-3-8B-Instruct on TL;DR for 4 epochs. Error bars over 10 seeds are reported.

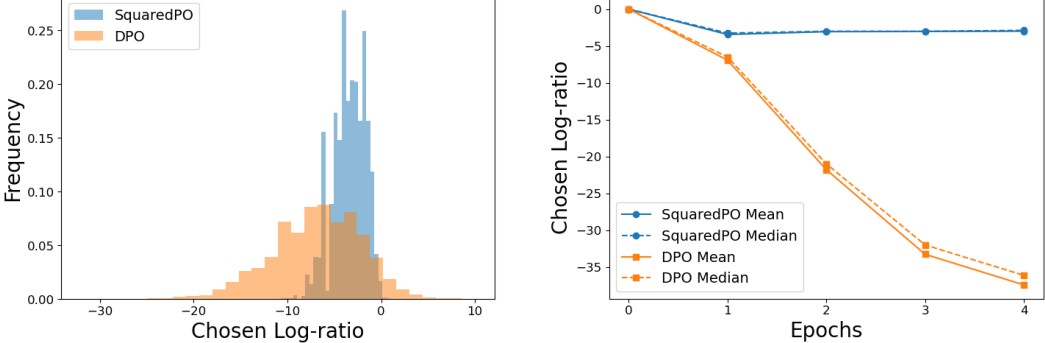

Figure 3: Left: histograms of chosen log-ratios $\log\left(\pi_\theta(y_w \mid x)/\pi_{\text{ref}}(y_w \mid x)\right)$ for all chosen responses in the training set $\mathcal{D}$ after one epoch of training. With SQUAREDPO, likelihood displacement is less severe; probabilities decrease less than under DPO. Right: the evolution of the mean and the median chosen log-ratio over epochs of training.

tion, we begin with the histograms in Figure 3 (left), which visualize the distribution of the values $\{\log\left(\pi_\theta\left(y_w \mid x\right)/\pi_{\text{ref}}\left(y_w \mid x\right)\right)\}_{(x,y_w,y_l)\in\mathcal{D}}$, dubbed 'chosen log-ratios'. This value is negative for winners that suffered from probability displacement. In Figure 3, $\pi_\theta$ is the model obtained after one training epoch. We see that SQUAREDPO is able to mitigate the likelihood displacement issue: while winner probabilities still decrease, the magnitude of the decrease is smaller. Notably, the most extreme decreases observed under DPO are absent in SQUAREDPO.

Beyond analyzing the snapshot after one epoch of training, we also examine how likelihood displacement evolves over the course of training. In Figure 6 in the appendix, we provide histograms for the chosen log-ratios of the models obtained from $1, 2, 3$ and $4$ epochs of alignment. These results indicate that the mitigation of extreme displacement by SQUAREDPO grows more pronounced with additional training. At an aggregate level, the right panel of Figure 3 presents the mean and median chosen log-ratios for each epoch, demonstrating that the decrease in DPO is by far more radical than in SQUAREDPO.

Lastly, we report an intriguing 'monotonicity' phenomenon we observe in the training dynamics of DPO: we find that in our experimental setting, nearly all ($99.63\%$) of the chosen responses whose probability decreased in the first epoch, continued decreasing monotonically in the three subsequent epochs. That is, for nearly all responses whose probability reduced from the reference checkpoint to the first epoch, the probability was also reduced from the first epoch to the second epoch and so on. While the literature contains many reports of the likelihood displacement phenomenon, we are, to the best of our knowledge, the first to report this monotonicity phenomenon, which is checked on a per-winner basis.

This phenomenon is substantially reduced under SQUAREDPO, as only $4.21\%$ of chosen responses that decreased in the first epoch continued to decrease monotonically thereafter. This empirical observation is in accordance with our mechanistic understanding of the SQUAREDPO loss (§4): when a winner $y_w$ suffers from likelihood displacement, its effective regularization coefficient $\beta_\theta(y_w, x)$ increases. This strengthens the regularization on that response and intuitively encourages its probability to return toward $\pi_{\text{ref}}(y_w \mid x)$, thereby breaking the downward monotonicity. See Figure 7 in the appendix for the evolution of log-ratios across training for ten individual winners, and Table 3 for additional details on monotonicity in several experimental settings.

## 6 RELATED WORK

The likelihood displacement phenomenon has been documented and analyzed in a growing body of work (Pang et al., 2024; Xiao et al., 2024; Shen et al., 2024; Nvidia et al., 2024; Wu et al., 2025; Pal et al., 2024; Rafailov et al., 2024; Fisch et al., 2025; D'Oosterlinck et al., 2025; Asadi et al., 2025; Razin et al., 2025). The connection between likelihood displacement and performance is discussed, e.g., in (Razin et al., 2025).

The use of $f$-divergences for direct preference optimization was recently introduced by Wang et al. (2024), who proposed the $f$-DPO loss. Our theoretical results in §3.1 are a non-trivial generalization of theirs. Perhaps closest to our work is Huang et al. (2025), who also analyze an instantiation of $f$-DPO with formal guarantees and robustness to over-optimization. While their focus is analyzing one specific instance in depth, we take a broader perspective and provide theoretical results that hold for general $f$.

Li et al. (2025) investigate the role of $f$-divergences within the RLVR framework, focusing on preserving solution diversity during online training. Other approaches, such as $f$-PO (Han et al., 2024) and $f$-DGP (Go et al., 2023), also use $f$-divergences for alignment, but do so by minimizing the divergence to a target model, rather than by replacing the KL term in the RLHF objective as in our work.

There are several works that propose losses that are similar in flavor to ours, however they are introduced heuristically rather than arise from theory. Similar to SQUAREDPO, SimPO (Meng et al., 2024) can also be viewed as DPO with adaptive $\beta$-s; however, in their case, the $\beta$-s depend only on the responses' length and are static throughout training. In $\beta$-DPO (Wu et al., 2024) and $\varepsilon$-DPO (Lee et al., 2025), the $\beta$-s, just as in our loss, depend both on the current model and on the specific prompt and response, however, they both, unlike us, introduce an additional hyperparameter.

Among the works that deal with likelihood displacement, our theoretical analysis most closely resembles that of Asadi et al. (2025). They, as well as others, e.g. (Pal et al., 2024; Wu et al., 2025), add an ad-hoc penalty term to the DPO loss to fight likelihood displacement, while we propose a one-term loss that arises from theory, and importantly does not add a hyperparameter, unlike the mentioned approaches that have the coefficient of the penalty term (typically $\lambda$) as an additional hyperparameter.

## 7 CONCLUSION

We characterized the set of DPO-inducing functions, i.e. functions $f$ that can be used in $f$-DPO, showing that convexity of $f$ is not a necessity. We studied the effect of the choice of $f$ on likelihood displacement, which is the phenomenon where during preference optimization both the rejected and the chosen responses decrease in probability, finding that $1 \leq \arg\min_{t \in \mathbb{R}_+} f(t)$ is a necessary condition to mitigate the decrease. We believe that the two properties we propose, namely displacement-resistant and DPO-inducing, may provide a general design principle for future methods in direct preference optimization.

With these two properties in mind, we analyze an example of a function that satisfies both desiderata, resulting in a novel loss we call SQUAREDPO. We find that this novel loss is not only competitive with DPO on standard benchmarks, but also offers a substantially greater degree of robustness to over-optimization and a clear empirical mitigation of likelihood displacement. Alongside our empirical examination of SQUAREDPO and DPO, we report a monotonicity phenomenon of DPO that is related to, but different from, likelihood displacement. We show that SQUAREDPO manages to alleviate this behavior of DPO as well.

**Limitations**  First, our experimental evaluation is restricted to a single dataset and model; although we provide additional displacement experiments with another model and another dataset in Appendix C.2. Second, the theoretical results regarding likelihood displacement focused on optimal solutions, abstracting away from optimizer behavior and training dynamics. Third, on the empirical side, we compared our method to DPO and $\chi$PO, while other DPO-style algorithms do exist. Another limitation is that all our experiments were conducted under LoRA finetuning. On the theoretical side, while our proposed method does mitigate displacement empirically, the displacement-resistance condition we introduce is formally proved to be necessary, though not established as sufficient. Identifying a sufficient condition is an important direction for future work. Finally, while our analysis applies to general choices of $f$, our experiments instantiate only one such function. Exploring other functions $f$ that satisfy our desiderata is a promising avenue for future research.

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

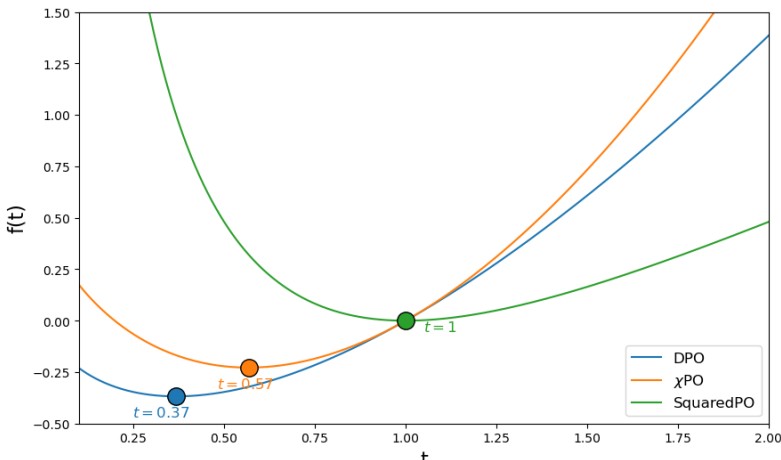

Figure 4: The $f$-divergence generators used by DPO ($t \log t$), $\chi$PO $\left(\frac{1}{2}(t-1)^2 + t \log t\right)$, and SQUAREDPO $\left(\frac{1}{2}(\log t)^2\right)$. The location of the global minimum of each function determines its susceptibility to likelihood displacement, with SQUAREDPO (minimum at $t = 1$) being the most resistant to displacement.

## A    COMPARISON TO RELATED PREFERENCE-OPTIMIZATION METHODS

In this appendix, we discuss in greater depth the connection between our results and several related methods mentioned in the related work.

The $\chi$PO loss suggested by Huang et al. (2025) is an instantiation of $f$-DPO with $f(t) = \frac{1}{2}(t-1)^2 + t \log t$[5]. This function attains a global minimum at $W(1) \approx 0.56714$, where $W$ is the Lambert $W$ function (Corless et al., 1996). Recall that our theory in § 3.2 suggests that whenever the $f$ used in $f$-DPO attains a global minimum at $c \in (0, 1]$, then, under mild conditions, the probabilities of all in-sample responses are expected to decrease by at least $c$ (Lemma 2). This suggests that if for two DPO-inducing functions $f_1$ and $f_2$ we have $\arg \min f_1 < \arg \min f_2 \leq 1$, then $f$-DPO with $f_1$ is more prone to likelihood displacement than $f$-DPO with $f_2$. Therefore, when comparing $\chi$PO with vanilla DPO, and recalling that the $f$ used by vanilla DPO, $f(t) = t \log t$, attains a global minimum at $e^{-1} = 0.36788$, we can expect $\chi$PO to be less prone to probability displacement than vanilla DPO (since $e^{-1} < W(1)$). This is in line with the findings of Huang et al. (2025), who analyze a setting in which DPO suffers from likelihood displacement while $\chi$PO does not (Remark B.1 therein). Regarding our method SQUAREDPO, which is $f$-DPO with the function $f(t) = \frac{1}{2}(\log t)^2$ that attains a minimum at $c = 1$, our theory suggests that it is less prone to likelihood displacement than both vanilla DPO and $\chi$PO. See Figure 4 for an illustration of the three generating functions $f$ mentioned in this paragraph.

## B    HYPERPARAMETERS AND IMPLEMENTATION DETAILS

**General training details**    Our implementation of the training pipeline is based on the DPO trainer from the TRL repository (von Werra et al., 2020). For all losses, we use a learning rate of $5e{-}7$, $\beta = 0.01$ (standard choices for DPO in this setting, e.g. (Meng et al., 2024)), a batch size of 64, a max sequence length of 2048, a linear learning rate scheduler, and the optimizer is AdamW (Loshchilov & Hutter, 2019). We use LoRA (Hu et al., 2022) with rank $r = 16$, scaling factor $\alpha = 32$, and dropout rate 0.05. Training was performed in `bf16` precision on 4×NVIDIA A100-SXM4-40GB GPUs. While we focus on LoRA fine-tuning for computational reasons, we expect the same empirical trends to hold under full fine-tuning.

---

[5]In general, the $\chi^2$ divergence is an $f$-divergence with $f(t) = \frac{1}{2}(t-1)^2$, however this function is not DPO-inducing and hence Huang et al. (2025) had to add the KL-like term $t \log t$.

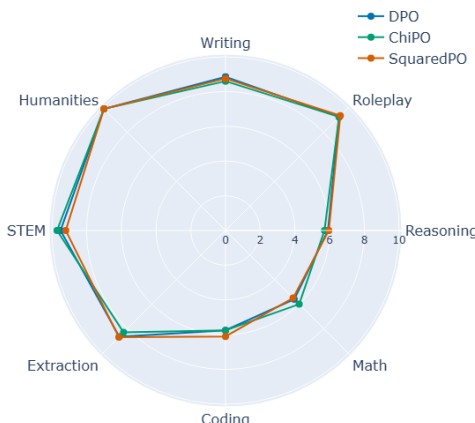

Figure 5: Category-level MT-Bench results for the models obtained from training Meta-Llama-3-8B-Instruct for one epoch on TL;DR using SQUAREDPO, $\chi$PO or DPO. For each of the eight MT-Bench categories, the reported value is the average score (across prompts in that category) assigned by an LLM-as-a-judge (gpt-4o). Scores are on a 0–10 scale, where higher is better.

**Clipping**   Recall that to calculate the SQUAREDPO loss for a sample $(x, y_w, y_l)$, we need to calculate $\beta_\theta (y, x) := \beta / \frac{\pi_\theta(y|x)}{\pi_{\mathrm{ref}}(y|x)}$ for any $y \in S_x$, that is, for $y_w$ and $y_l$. In the TRL repository, and in general, one works with log-probabilities rather than with raw probabilities. We calculate $\beta_\theta (y, x)$ by $\beta \exp\left(\min\left\{\log \pi_{\mathrm{ref}} (y \mid x) - \log \pi_\theta (y \mid x), 50\right\}\right)$. Apart from the clipping, this is equivalent to the original expression. The threshold for clipping, 50, is safe enough to prevent numerical issues and does not substantially change the learning curves based on some preliminary experiments we performed. We found it necessary to clip from above to prevent $\beta_\theta (y, x)$ from diverging to infinity in floating-point representation, whereas clipping from below was not necessary because $\beta_\theta (y, x)$ approaching zero is less problematic.

**Evaluation details**   To calculate win rates for the validation split, and to calculate win rates and length-controlled win rates for AlpacaEval 2, we use the alpaca_eval repository (Li et al., 2023). To calculate the MT-Bench results, we use the FastChat repository. In all settings, we generate responses to prompts by sampling with a temperature of $0.7$.

## C   ADDITIONAL EMPIRICAL RESULTS

### C.1   MT-BENCH RESULTS BY CATEGORY

Figure 5 presents a radar chart that breaks down the MT-Bench results from Table 2 into MT-Bench's eight categories. SQUAREDPO shows slight gains over DPO in coding tasks, performs worse on STEM tasks, and is otherwise largely on par.

### C.2   ADDITIONAL RESULTS ON LIKELIHOOD DISPLACEMENT

In Figure 6 we present histograms of the distribution of the log-ratios $\log(\pi_\theta(y_w \mid x)/\pi_{\mathrm{ref}}(y_w \mid x))$ for all the 92.9K chosen responses $y_w$ in the training split of TL;DR when using this dataset to perform preference optimization on Meta-Llama-3-8B-Instruct. The smaller the term $\log(\pi_\theta(y_w \mid x)/\pi_{\mathrm{ref}}(y_w \mid x))$ is, the more severe the likelihood displacement this winner witnessed. Notice how with DPO, already after one epoch a significant portion of the winners suffered from severe likelihood displacement, while SQUAREDPO manages to control the extent to which probabilities get reduced. As training progresses, under DPO we see the distribution shifting to more negative values, while SQUAREDPO remains relatively stable.

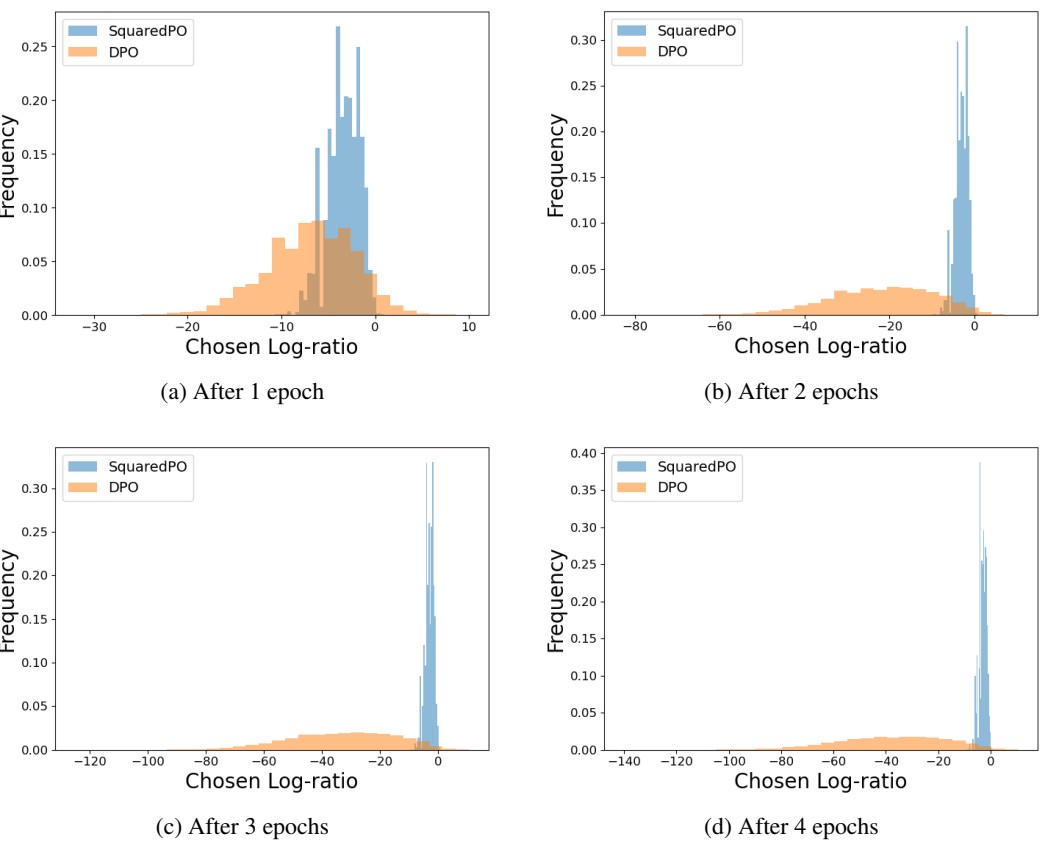

(a) After 1 epoch

(b) After 2 epochs

(c) After 3 epochs

(d) After 4 epochs

Figure 6: Histograms of chosen log-ratios $\log(\pi_\theta(y_w \mid x)/\pi_{\text{ref}}(y_w \mid x))$ for all chosen responses in the training set $\mathcal{D}$ at different stages of training. With SQUAREDPO, likelihood displacement is consistently less severe compared to DPO, and the difference becomes more pronounced as training progresses.

Table 3: Among the winners whose probability decreased after the first epoch, we report the percentage that continued decreasing in later epochs. Results are shown for DPO and SQUAREDPO in two settings: Llama + TL;DR and Qwen + UltraFeedback.

| | Llama + TL;DR | | Qwen + UltraFeedback | |
|---|---|---|---|---|
| **Condition** | **DPO (%)** | **SQUAREDPO (%)** | **DPO (%)** | **SQUAREDPO (%)** |
| Decrease from $1 \rightarrow 2$ | 99.99 | 10.91 | 93.72 | 25.56 |
| Monotone decrease $1 \rightarrow 3$ | 99.74 | 9.38 | 39.99 | 6.24 |
| Monotone decrease $1 \rightarrow 4$ | 99.63 | 4.21 | 35.57 | 1.64 |

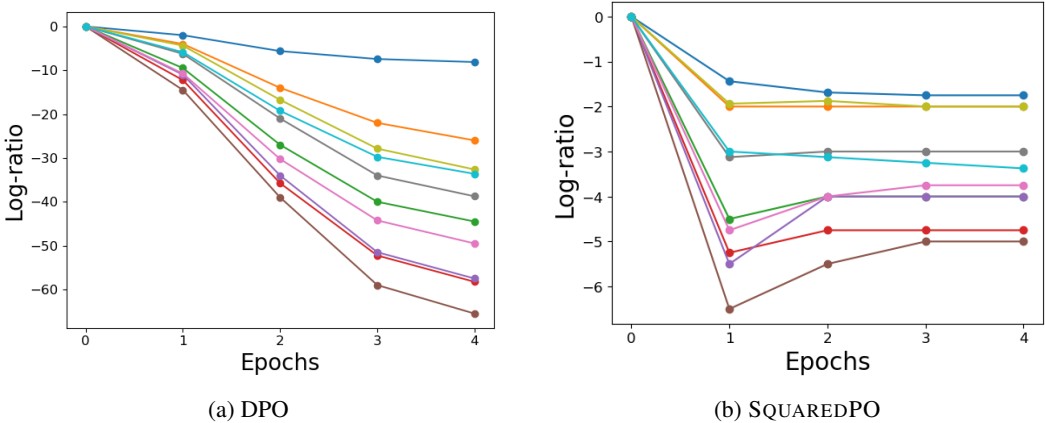

(a) DPO                    (b) SQUAREDPO

Figure 7: The evolution of the log-ratios $\log(\pi_\theta(y_w \mid x)/\pi_{\text{ref}}(y_w \mid x))$ during training for a random subset of 10 samples from the TL;DR training set. SQUAREDPO mitigates the 'monotonicity' phenomenon observed in DPO.

In Table 3 we provide additional results regarding the monotonicity phenomenon reported in §5.2. For example, when using the TL;DR dataset to finetune Meta-Llama-3-8B-Instruct with DPO, 99.99% of the winners whose probability decreased after the first epoch continued decreasing in the second, compared to only 10.91% under SQUAREDPO.

To diversify our experimental setting in the context of likelihood displacement across models and datasets, we additionally train Qwen2-0.5B-Instruct (Yang et al., 2024) on the UltraFeedback dataset (Cui et al., 2024) for four epochs. The empirical monotonicity analysis for this case is reported in Table 3. In this setting, 35.57% of winners who decreased under DPO in the first epoch continued to decrease monotonically until the fourth epoch, whereas under SQUAREDPO this proportion is merely 1.64%. Importantly, the trend that SQUAREDPO manages to break the downward monotonicity of winner probabilities compared to DPO holds consistently across all experimental settings. A mechanistic explanation of this trend is provided in the last paragraph of §5.2.

To further illustrate the trend in Table 3, we track the log-ratio trajectories of ten randomly selected winners from the TL;DR training set across the four training epochs of Meta-Llama-3-8B-Instruct under both DPO and SQUAREDPO, using the same winners for each method. We visualize this in Figure 7. Notice that under DPO, all ten winners display a monotonic decrease in probability, while for SQUAREDPO this is not the case. Moreover, notice the different scales in the $y$-axes; under SQUAREDPO, the decrease is of smaller magnitude.

# D    THEORETICAL RESULTS OMITTED FROM THE PAPER

## D.1    THEORETICAL RESULTS OMITTED FROM SUBSECTION 3.1

We start by defining the notion of an interior-inducing function, briefly mentioned in §3.1.

**Definition 3.** *We say a function $f : \mathbb{R}_+ \to \mathbb{R}$ is* interior-inducing *if for any reward model $r_\phi$, any reference policy $\pi_{ref}$, and any regularization coefficient $\beta > 0$, every optimal solution $\pi_\theta$ to the generalized RLHF problem* (5) *assigns strictly positive probability to all continuations, i.e., $\pi_\theta (y \mid x) > 0$ for all $x, y$.*

The condition in Definition 3 is equivalent to requiring that, for each prompt $x$, $\pi_\theta (\cdot \mid x)$ is in the relative interior (Beck, 2017) of the simplex, which motivates our terminology. Next, we establish that this definition is equivalent to the definition of a DPO-inducing function.

**Lemma 3.** *Let $f : \mathbb{R}_+ \to \mathbb{R}$ be a continuous function in $\mathbb{R}_+$, which is continuously differentiable in $\mathbb{R}_{++}$. Then, $f$ is DPO-inducing if and only if it is interior-inducing.*

*Proof.* Assume first that $f$ is not interior-inducing. Then, there exist $r_\phi, \pi_{\text{ref}}$, and $\beta$ such that an optimal solution to (5) satisfies $\pi_\theta (y_w \mid x) = 0$ for some $(x, y_w, y_l) \in \mathcal{D}$. Now, recall that for $f$ to be DPO-inducing, substituting optimal solutions to (5) into the BT model (1) should yield the $f$-DPO loss. However, to perform this substitution, one needs to express the rewards difference $r_\phi (x, y_w) - r_\phi (x, y_l)$ in terms of the optimal $\pi_\theta (y_w \mid x), \pi_\theta (y_l \mid x)$. Since we have $\pi_\theta (y_w \mid x) = 0$, this is impossible.

Assume that $f$ is interior-inducing. We need to show that $f$ is DPO-inducing, i.e., that substituting any optimal solution of problem (5) into the BT model (1) yields the $f$-DPO loss (6). Let $\pi_\theta$ be an optimal solution to (5). Since $f$ is interior-inducing, by definition $\pi_\theta (y_w \mid x) > 0$ and $\pi_\theta (y_l \mid x) > 0$ for any $(x, y_w, y_l) \in \mathcal{D}$. Now, we define

$$g(\pi_\theta) := \mathbb{E}_{y \sim \pi_\theta(\cdot|x)} [r_\phi (x, y)] - \beta D_f [\pi_\theta (\cdot \mid x) \| \pi_{\text{ref}} (\cdot \mid x)] \ ,$$

which is the objective function of problem (5). Using a technical result (see Lemma 4 below) implies

$$\frac{\partial g(\pi_\theta)}{\partial \pi_\theta (y_w \mid x)} = \frac{\partial g(\pi_\theta)}{\partial \pi_\theta (y_l \mid x)} \ ,$$

which reads as follows

$$r_\phi (x, y_w) - \beta f' \left( \frac{\pi_\theta (y_w \mid x)}{\pi_{\text{ref}} (y_w \mid x)} \right) = r_\phi (x, y_l) - \beta f' \left( \frac{\pi_\theta (y_l \mid x)}{\pi_{\text{ref}} (y_l \mid x)} \right) \ .$$

Rearranging the equation gives

$$r_\phi (x, y_w) - r_\phi (x, y_l) = \beta f' \left( \frac{\pi_\theta (y_w \mid x)}{\pi_{\text{ref}} (y_w \mid x)} \right) - \beta f' \left( \frac{\pi_\theta (y_l \mid x)}{\pi_{\text{ref}} (y_l \mid x)} \right) \ ,$$

and substituting this difference into the BT model (1) yields the $f$-DPO loss (6), as required. $\qquad \square$

We next state Theorem 1, which provides a full characterization of DPO-inducing functions.

**Theorem 1.** *Let $f : \mathbb{R}_+ \to \mathbb{R}$ be a continuous function in $\mathbb{R}_+$, which is continuously differentiable in $\mathbb{R}_{++}$.*

    *i If $f (0) = \infty$, then $f$ is DPO-inducing.*

    *ii Suppose $f (0) < \infty$. Then, $f$ is DPO-inducing if and only if the ratio $\frac{f(t) - f(0)}{t}$ is not bounded from below in any right neighborhood of $0$ (i.e., $\forall a > 0, \forall M \in \mathbb{R}, \exists t \in (0, a)$ such that $\frac{f(t) - f(0)}{t} < M$).*

The proof for Theorem 1 can be found in Appendix E.1. Recall that in Corollary 1 we add an assumption under which the condition of the characterization becomes easier to check.

## D.2   THE OPTIMAL SOLUTIONS OF PROBLEM (7)

In this appendix, we provide and briefly discuss the optimal solution(s) to Problem 7 under the additional assumption that $f$ is convex, differentiable, and $f'$ is invertible. We do not prove it here, but rather rephrase it in simpler notations and prove it in Lemma 5.

First, notice that Problem 7 is separable over different $x$-s and hence we can provide the optimal solution $\pi_\theta$ separately for each $x \in \mathcal{D}$. So let $x \in \mathcal{D}$. Denote $\hat{r}_x := \max_{y \notin S_x} r_\phi(x, y)$. If the condition

$$\sum_{y \in S_x} \pi_{\text{ref}}(y \mid x) f'^{-1}\left(\frac{r_\phi(x, y) - \hat{r}_x}{\beta}\right) < 1 \tag{10}$$

holds, then the set of optimal solutions $\pi_\theta(\cdot \mid x)$ is the set of $\pi_\theta(\cdot \mid x)$ that satisfy

$$\begin{cases} \pi_\theta(y \mid x) = \pi_{\text{ref}}(y \mid x) f'^{-1}\left(\frac{r_\phi(x,y) - \hat{r}_x}{\beta}\right), & \forall y \in S_x, \\ \pi_\theta(y \mid x) = 0, & \forall y \notin S_x,\ r_\phi(x, y) < \hat{r}_x \\ \sum_{\substack{y \notin S_x \\ r_\phi(x,y) = \hat{r}_x}} \pi_\theta(y \mid x) = 1 - \sum_{y \in S_x} \pi_{\text{ref}}(y \mid x) f'^{-1}\left(\frac{r_\phi(x,y) - \hat{r}_x}{\beta}\right). \end{cases} \tag{11}$$

Otherwise, the unique optimal solution is:

$$\begin{cases} \pi_\theta(y \mid x) = \pi_{\text{ref}}(y \mid x) f'^{-1}\left(\frac{r_\phi(x,y) + \mu}{\beta}\right), & \forall y \in S_x, \\ \pi_\theta(y \mid x) = 0, & \forall y \notin S_x. \end{cases} \tag{12}$$

where $\mu$ is a solution to $\sum_{y \in S_x} \pi_{\text{ref}}(y \mid x) f'^{-1}\left(\frac{r_\phi(x,y) + \mu}{\beta}\right) = 1$.

This wraps up the characterization of the optimal set of the weird optimization problem (7). Notice that in both cases, the optimal solutions can assign zero probability to out-of-sample responses. This demonstrates that Problem (7) does not guardrail out-of-sample responses to stay close in probability to $\pi_{\text{ref}}$. We rephrase and prove this characterization with more convenient notations in Lemma 5. We note that this result appears in Asadi et al. (2025) for the special case $f_{\text{KL}}(t) = t \log t$.

# E    PROOFS

## E.1    PROOF OF THEOREM 1

*Proof.* For simplicity of the proof, we define

$$\max_{\mathbf{p} \in \Delta^n} g(\mathbf{p}) := \left\{ \mathbf{r}^T \mathbf{p} - \beta D_f(\mathbf{p} \parallel \mathbf{q}) \right\}. \tag{13}$$

We first prove the first item. Assume that $f(0) = \infty$, and we will show that any optimal solution $\mathbf{p}$ must be in $\Delta_{++}^n = \{\mathbf{p} \in \Delta^n \mid \mathbf{p} > 0\}$. Indeed, any vector $\mathbf{p}$, for which $p_i = 0$ for some $i \in [n]$, we have that $g(\mathbf{p}) = -\infty$. This obviously implies that such vectors $\mathbf{p}$ cannot be maximizers of $g$. Therefore, we get that any optimal solution lies in $\Delta_{++}^n$.

We now prove the second item. Assume that $f(0) < \infty$.

$\Longrightarrow$: We assume on the way of contradiction that the ratio $t^{-1}[f(t) - f(0)]$ is bounded from below. We will prove that $f$ is not interior-inducing by constructing an instance of the problem (13) with specific $\mathbf{r}$ and $\mathbf{q}$ such that $\mathbf{p}^* = \mathbf{e_1}$ is an optimal solution.

To this end, since the ratio $t^{-1}[f(t) - f(0)]$ is bounded from below, there exist $M \in \mathbb{R}$ and $a > 0$ such that for all $t \in (0, a)$ we have $t^{-1}[f(t) - f(0)] \geq M$. Since $t^{-1}[f(t) - f(0)]$ is continuous in $[a, n]$, by the Extreme Value Theorem there exists $K_1$ such that $t^{-1}[f(t) - f(0)] \geq K_1$ for all $t \in [a, n]$. Denoting $m = \min\{K_1, M\}$, we get $t^{-1}[f(t) - f(0)] \geq m$ for all $t \in (0, n]$. Moreover, let $D = \max_{t \in [1, 2n]} f'(t)$, which is finite by the assumption that $f$ is continuously differentiable in $\mathbb{R}_{++}$ and by the Extreme Value Theorem.

Now, we are ready to define $\mathbf{r}$ and $\mathbf{q}$. We consider problem (13) with $q_i = 1/n$ for all $i \in [n]$, $r_1 = \max\{\beta D - \beta m + 1, 1\}$, and $r_i = 0$ for all $i \geq 2$. We will show that $\mathbf{p}^* = \mathbf{e_1}$ is an optimal solution. To this end, since the problem obviously must have a maximizer, it is enough to show that no $\mathbf{p}$ with $p_i < 1$ can be an optimal solution.

We take $\mathbf{p} \in \Delta_n$ with $p_1 < 1$ such that $p_i > 0$ for some $i \in [n]$. Let $\mathbf{p}' = \mathbf{p} + p_i (\mathbf{e_1} - \mathbf{e_i})$, that is, the probability vector obtained by moving all the mass from $i$ to 1.

Then, from the definition of $\mathbf{p}'$, we get

$$
g(\mathbf{p}') - g(\mathbf{p}) = r_1 (p_1 + p_i) - \beta \sum_{\ell=1}^{n} \frac{1}{n} f(np'_\ell) - r_1 p_1 + \beta \sum_{\ell=1}^{n} \frac{1}{n} f(np_\ell)
$$
$$
= r_1 p_i - \frac{\beta}{n} f(n(p_1 + p_i)) - \frac{\beta}{n} f(0) + \frac{\beta}{n} f(np_1) + \frac{\beta}{n} f(np_i)
$$
$$
= r_1 p_i + \frac{\beta}{n} (f(np_i) - f(0)) - \frac{\beta}{n} (f(np_1 + np_i) - f(np_1)) .
$$

Dividing by $p_i > 0$ we get

$$
\frac{g(\mathbf{p}') - g(\mathbf{p})}{p_i} = r_1 + \beta \frac{f(np_i) - f(0)}{np_i} - \beta \frac{f(np_1 + np_i) - f(np_1)}{np_i}
$$
$$
\geq r_1 + \beta m - \beta \frac{f(np_1 + np_i) - f(np_1)}{np_i} , \tag{14}
$$

where the inequality is because $np_i \in (0, n]$.

Now, to bound the last term, first we notice that if $p_1 < 1/n$, then $\mathbf{p}$ is not optimal. Indeed, in this case there is necessarily $j \neq 1$ such that $p_j > 1/n$. Therefore, by considering the value of $g$ on the vector where $p_j$ and $p_1$ switches, we get higher value than $g(\mathbf{p})$ since $r_1 \geq 1 > 0 = r_j$.

Therefore, we can assume $p_1 \geq 1/n$. Moreover, by Lagrange's mean value theorem, there exists a $c \in (np_1, np_1 + np_i)$ such that

$$
\frac{f(np_1 + np_i) - f(np_1)}{np_i} = f'(c) \leq \max_{t \in [1, 2n]} f'(t) = D
$$

where the inequality follows from the fact that $c \in [1, 2n]$. Indeed, since $p_1 \geq 1/n$, we have $1 \leq np_1$. Also, $np_1 + np_i \leq n$ since $p_1 + p_i \leq 1$. Therefore, $c \in [1, n]$.

Using this in (14), we get

$$
\frac{g(\mathbf{p}') - g(\mathbf{p})}{p_i} \geq r_1 + \beta m - \beta D \geq \beta D - \beta m + 1 + \beta m - \beta D = 1 > 0 ,
$$

where the second inequality follows from the fact that $r_1 \geq \beta D - \beta m + 1$. Therefore,

$$
g(\mathbf{p}') - g(\mathbf{p}) > 0 , \tag{15}
$$

which shows that $\mathbf{p}$ is not optimal, as required.

$\Longleftarrow$: Assume that $f(0) < \infty$ and that the ratio $t^{-1}[f(t) - f(0)]$ is not bounded from below, and we will prove that $f$ is interior-inducing.

Let $\mathbf{r} \in \mathbb{R}^n, \mathbf{q} \in \Delta_{++}^n$, and $\beta > 0$. We need to show that any optimal solution to

$$
\max_{\mathbf{p} \in \Delta^n} g(\mathbf{p}) := \{\mathbf{r}^T \mathbf{p} - \beta D_f(\mathbf{p} \| \mathbf{q})\}
$$

satisfies $\mathbf{p} \in \Delta_{++}^n$. We assume in contradiction that $p_i = 0$ for some $i \in [n]$ and we will prove that $\mathbf{p}$ is not an optimal solution.

Let $j \in [n]$ be such that $p_j > 0$, and define $\mathbf{p}^\varepsilon = \mathbf{p} + \varepsilon (\mathbf{e_i} - \mathbf{e_j})$. Notice that for every $\varepsilon < p_j$, we indeed have $\mathbf{p}^\varepsilon \in \Delta^n$. Then, for all $\varepsilon < p_j$, we have

$$g\left(\mathbf{p}^{\varepsilon}\right) - g\left(\mathbf{p}\right) = \mathbf{r}^{T}\left(\mathbf{p}^{\varepsilon} - \mathbf{p}\right) - \beta\sum_{\ell=1}^{n} q_{\ell}f\left(\frac{p_{\ell}^{\varepsilon}}{q_{\ell}}\right) + \beta\sum_{\ell=1}^{n} q_{\ell}f\left(\frac{p_{\ell}}{q_{\ell}}\right)$$

$$= \varepsilon\mathbf{r}^{T}\left(\mathbf{e_i} - \mathbf{e_j}\right) - \beta\left(q_if\left(\frac{p_i^{\varepsilon}}{q_i}\right) + q_jf\left(\frac{p_j^{\varepsilon}}{q_j}\right)\right) + \beta\left(q_if\left(\frac{p_i}{q_i}\right) + q_jf\left(\frac{p_j}{q_j}\right)\right)$$

$$= \varepsilon\left(r_i - r_j\right) - \beta q_i\left(f\left(\frac{\varepsilon}{q_i}\right) - f\left(0\right)\right) + \beta q_j\left(f\left(\frac{p_j}{q_j}\right) - f\left(\frac{p_j - \varepsilon}{q_j}\right)\right). \quad (16)$$

We now wish to find $\varepsilon$ such that $g\left(\mathbf{p}^{\varepsilon}\right) - g\left(\mathbf{p}\right) > 0$, which will prove that $\mathbf{p}$ is not an optimal solution. Note that

$$\lim_{\varepsilon\to 0^+} \frac{q_j\left(f\left(\frac{p_j}{q_j}\right) - f\left(\frac{p_j-\varepsilon}{q_j}\right)\right)}{\varepsilon} = \lim_{\varepsilon\to 0^+} \frac{f\left(\frac{p_j}{q_j} - \frac{\varepsilon}{q_j}\right) - f\left(\frac{p_j}{q_j}\right)}{-\frac{\varepsilon}{q_j}} = f'\left(\frac{p_j}{q_j}\right) \in \mathbb{R}\,,$$

since $f$ is differentiable at $\mathbb{R}_{++}$. Therefore, by the definition of the limit, there exists $\bar{\varepsilon} > 0$ such that for all $\varepsilon \in (0, \bar{\varepsilon})$,

$$\left|\frac{q_j\left(f\left(\frac{p_j}{q_j}\right) - f\left(\frac{p_j-\varepsilon}{q_j}\right)\right)}{\varepsilon} - f'\left(\frac{p_j}{q_j}\right)\right| < 1\,,$$

which means that

$$\frac{q_j\left(f\left(\frac{p_j}{q_j}\right) - f\left(\frac{p_j-\varepsilon}{q_j}\right)\right)}{\varepsilon} > f'\left(\frac{p_j}{q_j}\right) - 1\,. \quad (17)$$

Considering the ratio condition with $M = f'\left(\frac{p_j}{q_j}\right) + \frac{1}{\beta}\left(r_j - r_i\right) - 2$ and $a = \min\left\{\bar{\varepsilon}, p_j\right\}/q_i$, we get that there exists $t \in (0, a)$ such that $\frac{f(t)-f(0)}{t} < M$. Defining $\varepsilon_0 = tq_i$, we then get $\varepsilon_0 \in (0, \bar{\varepsilon})$ and $\varepsilon_0 \in (0, p_j)$. Thus,

$$\frac{f\left(\frac{\varepsilon_0}{q_i}\right) - f\left(0\right)}{\frac{\varepsilon_0}{q_i}} = \frac{f\left(t\right) - f\left(0\right)}{t} < M\,. \quad (18)$$

Therefore, by combining from (16), (17), and (18) we get

$$g\left(\mathbf{p}^{\varepsilon_0}\right) - g\left(\mathbf{p}\right) > \varepsilon_0\left(r_i - r_j\right) - \beta M\varepsilon_0 + \beta\varepsilon_0\left(f'\left(\frac{p_j}{q_j}\right) - 1\right) = \beta\varepsilon_0 > 0\,,$$

where the last equality follows from the definition of $M$. This shows that $g\left(\mathbf{p}^{\varepsilon_0}\right) > g\left(\mathbf{p}\right)$, which contradicts the optimality of $\mathbf{p}$. □

### E.2 PROOF OF COROLLARY 1

*Proof.* Since $\lim_{t\to 0^+} f'\left(t\right)$ exists and $f'$ is continuous, there is $L$ such that

$$\lim_{t\to 0^+} \frac{f\left(t\right) - f\left(0\right)}{t} = f'\left(0\right) = \lim_{t\to 0^+} f'\left(t\right) = L. \quad (19)$$

Therefore, $L = -\infty$ if and only if the ratio $t^{-1}[f\left(t\right) - f\left(0\right)]$ is not bounded from below. This concludes the result as an immediate consequence of Theorem 1. □

### E.3 PROOF OF LEMMA 1

*Proof.* Since $f$ is DPO-inducing, it satisfies the conditions in Theorem 1. Therefore, a straightforward generalization of our proof of Theorem 1 can show that any optimal solution $\pi_\theta$ to Problem (7) satisfies $\pi_\theta\left(y \mid x\right) > 0$ for any $y \in S_x$. Therefore, for any $(x, y_w, y_l) \in \mathcal{D}$ we have

$\pi_\theta\left(y_w \mid x\right) > 0$ and $\pi_\theta\left(y_l \mid x\right) > 0$. This implies, by Technical Lemma 4, the following equality of derivatives:

$$\frac{\partial g(\pi_\theta)}{\partial \pi_\theta\left(y_w \mid x\right)} = \frac{\partial g(\pi_\theta)}{\partial \pi_\theta\left(y_l \mid x\right)} , \tag{20}$$

where we denote by $g\left(\pi_\theta\right)$ the objective function of (7). By calculation, this gives us

$$r_\phi\left(x, y_w\right) - r_\phi\left(x, y_l\right) = \beta f'\left(\frac{\pi_\theta\left(y_w \mid x\right)}{\pi_{\text{ref}}\left(y_w \mid x\right)}\right) - \beta f'\left(\frac{\pi_\theta\left(y_l \mid x\right)}{\pi_{\text{ref}}\left(y_l \mid x\right)}\right) , \tag{21}$$

which when substituted into the BT model (1) yields the $f$-DPO loss (6), as required. $\square$

### E.4 PROOF OF LEMMA 2

*Proof.* Suppose in contradiction that there exists an optimal solution $\pi_\theta$ to Problem (7) such that for some $x$ and for some $y \in S_x$, $\pi_\theta\left(y \mid x\right) > c \cdot \pi_{\text{ref}}\left(y \mid x\right)$. Denote by $\hat{y} \in \arg\max_{y' \notin S_x} r_\phi\left(x, y'\right)$ an out-of-sample response for $x$ with a maximal reward. We define

$$\pi'_\theta\left(z \mid x\right) = \begin{cases} c \cdot \pi_{\text{ref}}\left(y \mid x\right), & z = y, \\ \pi_\theta\left(\hat{y} \mid x\right) + \pi_\theta\left(y \mid x\right) - \pi'_\theta\left(y \mid x\right), & z = \hat{y} \\ \pi_\theta\left(z \mid x\right), & z \notin \{y, \hat{y}\}. \end{cases}$$

Since $c \in (0, 1]$, we immediately obtain that $\pi'_\theta\left(\cdot \mid x\right)$ is indeed a valid probability distribution. From (7) and using the definition of $\pi'_\theta\left(z \mid x\right)$, we have

$$\begin{aligned} g\left(\pi'_\theta\right) - g\left(\pi_\theta\right) &= \pi'_\theta\left(\hat{y} \mid x\right) r_\phi\left(x, \hat{y}\right) + \pi'_\theta\left(y \mid x\right) r_\phi\left(x, y\right) - \beta\pi_{\text{ref}}\left(y \mid x\right) f\left(\frac{\pi'_\theta\left(y \mid x\right)}{\pi_{\text{ref}}\left(y \mid x\right)}\right) \\ &\quad - \left(\pi_\theta\left(\hat{y} \mid x\right) r_\phi\left(x, \hat{y}\right) + \pi_\theta\left(y \mid x\right) r_\phi\left(x, y\right) - \beta\pi_{\text{ref}}\left(y \mid x\right) f\left(\frac{\pi_\theta\left(y \mid x\right)}{\pi_{\text{ref}}\left(y \mid x\right)}\right)\right) \\ &= r_\phi\left(x, \hat{y}\right)\left(\pi_\theta\left(y \mid x\right) - \pi'_\theta\left(y \mid x\right)\right) - r_\phi\left(x, y\right)\left(\pi_\theta\left(y \mid x\right) - \pi'_\theta\left(y \mid x\right)\right) \\ &\quad - \beta\pi_{\text{ref}}\left(y \mid x\right) f\left(c\right) + \beta\pi_{\text{ref}}\left(y \mid x\right) f\left(\frac{\pi_\theta\left(y \mid x\right)}{\pi_{\text{ref}}\left(y \mid x\right)}\right) . \end{aligned}$$

Now, since $c$ is the unique global minimum of $f$, we have $f\left(c\right) < f\left(\frac{\pi_\theta(y|x)}{\pi_{\text{ref}}(y|x)}\right)$, and by our assumption $\pi_{\text{ref}} > 0$, which implies that

$$g\left(\pi'_\theta\right) - g\left(\pi_\theta\right) > r_\phi\left(x, \hat{y}\right)\left(\pi_\theta\left(y \mid x\right) - \pi'_\theta\left(y \mid x\right)\right) - r_\phi\left(x, y\right)\left(\pi_\theta\left(y \mid x\right) - \pi'_\theta\left(y \mid x\right)\right) .$$

Now, recalling that $r_\phi\left(x, \hat{y}\right) = \max_{y' \notin S_x} r_\phi\left(x, y'\right) \geq r_\phi\left(x, y\right)$ by the assumption of the Lemma and that $\pi_\theta\left(y \mid x\right) > \pi'_\theta\left(y \mid x\right)$ by construction of $\pi'_\theta$, we get that $g\left(\pi'_\theta\right) - g\left(\pi_\theta\right) > 0$ in contradiction to the optimality of $\pi_\theta$. $\square$

## F TECHNICAL LEMMAS AND THEIR PROOFS

We denote and define the $n - 1$ dimensional simplex by $\Delta^n = \left\{\mathbf{p} \in \mathbb{R}^n_+ \mid \sum_{i=1}^n p_i = 1\right\}$.

**Lemma 4.** *Let $g : \Delta^n \to \mathbb{R}$ be a function and let $\mathbf{p}^*$ be an optimal solution of the problem*

$$\max_{\mathbf{p} \in \Delta^n} g\left(\mathbf{p}\right) . \tag{22}$$

*Suppose that $g$ is differentiable at $\mathbf{p}^*$. Then, for any $i, j \in [n]$ for which $p_i^*, p_j^* > 0$, we have $\frac{\partial g(\mathbf{p}^*)}{\partial p_i} = \frac{\partial g(\mathbf{p}^*)}{\partial p_j}$.*

*Proof.* Suppose in contradiction $\frac{\partial g(\mathbf{p}^*)}{\partial p_i} < \frac{\partial g(\mathbf{p}^*)}{\partial p_j}$. Let $\mathbf{v} = \mathbf{e}_j - \mathbf{e}_i$, where $\mathbf{e}_i$ and $\mathbf{e}_j$ are the $i$-th and $j$-th standard basis vectors in $\mathbb{R}^n$. Then, we have

$$\mathbf{v}^T \nabla g\left(\mathbf{p}^*\right) = \frac{\partial g\left(\mathbf{p}^*\right)}{\partial p_j} - \frac{\partial g\left(\mathbf{p}^*\right)}{\partial p_i} > 0 .$$

Since $g$ is differentiable at $\mathbf{p}^*$, using the definition of the directional derivative of $g$ along the vector $\mathbf{v}$ (see Section 1.5.1 in Beck (2014)), we get that

$$0 < \mathbf{v}^T \nabla g\left(\mathbf{p}^*\right) = \lim_{h \to 0} \frac{g\left(\mathbf{p}^* + h\mathbf{v}\right) - g\left(\mathbf{p}^*\right)}{h} .$$

Therefore, for a small enough $h > 0$, we have

$$g\left(\mathbf{p}^* + h\mathbf{v}\right) > g\left(\mathbf{p}^*\right) .$$

Furthermore, since $p_i^*, p_j^* > 0$, there exsits a small enough $h > 0$ such that $\mathbf{p}^* + h\mathbf{v} \in \Delta^n$. These two facts contradict the optimality of $\mathbf{p}^*$. Since similar arguments will derive a contradiction in the case where $\frac{\partial g(\mathbf{p}^*)}{\partial p_i} > \frac{\partial g(\mathbf{p}^*)}{\partial p_j}$, we obtain the desired result. $\qquad\square$

**Lemma 5.** *Let $f : \mathbb{R}_+ \to \mathbb{R}$ be a convex DPO-inducing function, which is differentiable and $f'$ is invertible. Let $S$ be a strict subset of $[n] := \{1, \ldots, n\}$. For given $\mathbf{r} \in \mathbb{R}^n$, $\mathbf{q} \in \Delta^n$, $\beta > 0$, we consider the following problem:*

$$\max_{\mathbf{p} \in \Delta^n} \left\{ \sum_{i=1}^n p_i r_i - \beta \sum_{i \in S} q_i f\left(p_i / q_i\right) \right\} . \tag{23}$$

*Denote $\hat{r} = \max_{i \notin S} r_i$ and $z_i = q_i f'^{-1}\left((r_i - \hat{r})/\beta\right)$ for any $i \in S$. If $\sum_{i \in S} z_i < 1$, then the set of optimal solutions to* (23) *can be described as all vectors $p$ satisfying*

$$\begin{cases} p_i = z_i, & \forall i \in S, \\ p_i = 0, & \forall i \notin S,\ r_i < \hat{r} \\ \sum_{\substack{i \notin S \\ r_i = \hat{r}}} p_i = 1 - \sum_{i \in S} z_i. \end{cases} \tag{24}$$

*Otherwise, the unique optimal solution to* (23) *is:*

$$\begin{cases} p_i = q_i f'^{-1}\left(\frac{r_i + \mu}{\beta}\right), & \forall i \in S, \\ p_i = 0, & \forall i \notin S. \end{cases} \tag{25}$$

*where $\mu$ is a solution to $\sum_{i \in S} q_i f'^{-1}\left((r_i + \mu)/\beta\right) = 1$.*

*Proof.* Since the problem is convex (since $f$ is convex) and Slater's condition holds, the set of optimal solutions coincides with the set of KKT points. The Lagrangian in this case is given by

$$L\left(\mathbf{p}, \boldsymbol{\lambda}, \mu\right) = \langle \mathbf{r}, \mathbf{p} \rangle - \beta \sum_{i \in S} q_i f\left(\frac{p_i}{q_i}\right) + \langle \boldsymbol{\lambda}, \mathbf{p} \rangle + \mu \left(\sum_{i=1}^n p_i - 1\right) , \tag{26}$$

where $\boldsymbol{\lambda} \in \mathbb{R}_+^n$ and $\mu \in \mathbb{R}$. The KKT conditions are

$$\begin{cases} \frac{\partial L}{\partial p_i} = r_i - \beta f'\left(\frac{p_i}{q_i}\right) + \lambda_i + \mu = 0, & \forall i \in S, \\ \frac{\partial L}{\partial p_i} = r_i + \lambda_i + \mu = 0, & \forall i \notin S, \\ p_i \geq 0, \quad \sum_{i=1}^n p_i = 1, & \text{(feasibility)}, \\ \lambda_i \geq 0, & \forall i \in [n], \quad \text{(multipliers)}, \\ \lambda_i p_i = 0, & \forall i \in [n], \quad \text{(complementary slackness)}. \end{cases}$$

Since we assume that $f$ is DPO-inducing, by Lemma 3 we get that it is also interior-inducing, and therefore any solution $\mathbf{p}$ to (23) satisfies $p_i > 0$ for all $i \in S$. Therefore $\lambda_i = 0$ for all $i \in S$, and the KKT conditions can be equivalently written as:

$$
\begin{cases}
p_i = q_i f'^{-1}\left(\frac{r_i+\mu}{\beta}\right), & \forall i \in S. \\
r_i + \lambda_i + \mu = 0, & \forall i \notin S, \\
\mathbf{p} \in \Delta^n, & \text{(feasibility)}, \\
\lambda_i \geq 0, & \forall i \notin S, \quad \text{(multipliers)}, \\
\lambda_i p_i = 0, & \forall i \notin S, \quad \text{(complementary slackness)}.
\end{cases}
$$

We now split the proof into the two cases as mentioned in the lemma. Recall that $\hat{r} = \max_{i \notin S} r_i$ and $z_i = q_i f'^{-1}\left((r_i - \hat{r})/\beta\right)$.

- Assume that $\sum_{i \in S} z_i < 1$. First, the points described in (24) are all KKT points with $\mu = -\hat{r}$ and

$$
\begin{cases}
\lambda_i = 0, & \forall i \notin S,\ r_i = \hat{r}, \\
\lambda_i = \hat{r} - r_i, & \forall i \notin S,\ r_i < \hat{r}.
\end{cases}
$$

It remains to show that no other KKT points exist. To this end, let $\mathbf{p}$ be any KKT point. We will prove that $\mathbf{p}$ coincides with one of the solutions in (24). By feasibility, it suffices to establish that $p_i = z_i$ for all $i \in S$ and $p_i = 0$ for all $i \notin S$ such that $r_i < \hat{r}$.

For any $i \notin S$, we have that $r_i + \lambda_i + \mu = 0$. Since $\lambda_i \geq 0$, we have that $\mu \leq -r_i$ for all $i \notin S$ and therefore also $\mu \leq -\hat{r}$. If $p_i = 0$ for all $i \notin S$, since $\sum_{i \in S} z_i < 1$, we obtain a contradiction since

$$
1 = \sum_{i \in S} p_i = \sum_{i \in S} q_i f'^{-1}\left(\frac{r_i + \mu}{\beta}\right) \leq \sum_{i \in S} q_i f'^{-1}\left(\frac{r_i - \hat{r}}{\beta}\right) = \sum_{i \in S} z_i < 1,
$$

where the first inequality is because $f'^{-1}$ is increasing (by convexity of $f$) and since $\mu \leq -\hat{r}$.

Hence some $p_i > 0$ for $i \notin S$. Complementary slackness implies $\lambda_i = 0$ and thus $r_i = -\mu$. Moreover, $r_i = r_j + \lambda_j$ for all $j \notin S$, since $r_j + \lambda_j = -\mu$. Since $\lambda_j \geq 0$, we have that $r_i \geq r_j$ for all $j \notin S$, which means that $r_i = \hat{r}$. Hence $\mu = -\hat{r}$, which means that $p_i = z_i$ as desired.

- Assume that $\sum_{i \in S} z_i \geq 1$. In this case, we prove that the unique KKT point is the point given in (25). Note that by feasibility it is enough to show that $p_i = 0$ for all $i \notin S$.

Suppose, in contradiction, that $p_\ell > 0$ for some $\ell \notin S$. Then, $\lambda_\ell = 0$ and hence $\mu = -r_\ell$. Moreover,

$$
1 \geq p_\ell + \sum_{i \in S} q_i f'^{-1}\left(\frac{r_i - r_\ell}{\beta}\right) \geq p_\ell + \sum_{i \in S} q_i f'^{-1}\left(\frac{r_i - \hat{r}}{\beta}\right) \geq p_\ell + 1 > 1, \quad (27)
$$

where the first inequality is by feasibility of $\mathbf{p}$, the second inequality is since $f'^{-1}$ is increasing (by convexity of $f$) and $r_\ell \leq \hat{r}$, and the penultimate inequality is a rearrangement of the assumption of Case II (recall that $z_i = q_i f'^{-1}\left((r_i - \hat{r})/\beta\right)$).

$\square$

