# OpenReview forum: "Displacement-Resistant Extensions of DPO with Nonconvex $f$-Divergences"
_ICLR.cc/2026/Conference — ICLR 2026 Poster_

### Official Review · Reviewer_Y1Xo · 2025-10-24

**Soundness:** 4
**Presentation:** 3
**Contribution:** 2
**Rating:** 6
**Confidence:** 2

**Summary:**

This paper studies a generalization of $f$-DPO to non-convex $f$ generator functions, and identifies a simple condition to enforce on the generator function $f$ to be more resistant to the phenomenon of "likelihood displacement" affecting DPO, by which the probabilities of chosen responses in the preference dataset tend to 0 as the training progresses. The identified condition is that the minimum of $f(t)$ should occur at $t \geq 1$. They call such functions "displacement-resistant".  The paper proposes a specific generator function that they term SquaredPO and compares it against DPO on TL;DR.

**Strengths:**

- S1) The paper discusses novel ideas on the realm of using $f$ divergences as penalty terms in KL-regularized RL, specifically in the context of $f$-DPO.
- S2) It comes with an extensive theoretical analysis of the discussed algorithms

**Weaknesses:**

- W1) The paper, while clear and well-written, was also a bit terse to read for me. This is highly subjective, and I don't weigh this weakness strongly in my evaluation.
- W2) The usage of a non-convex $f$ felt quite arbitrary, and there is no comparison with a convex displacement-resistant function. In a sense, the generalization to non-convex $f$ might be interesting, but it's unclear from the content of this paper what can we gain from it.
- W3) More in general, the comparison of the proposed technique is very limited as it only considers one baseline (DPO) and one dataset (TL;DR) , as noted by the authors in the limitations section.  While the authors claim that they leave further comparisons for future work, I don't think this is an aspect that can be relegated to future work.

**Questions:**

- Q1) Your "displacement-resistant" condition is a necessary, but is it sufficient to avoid likelihood displacement? If not, isn't the name "displacement-resistant" a bit misleading?
- Q2) Your displacement-resistant is quite simple. Wouldn't it predict that a simple change of variables, say $t' = t - (1- e^{-1})$ for the reverse KL, work just as well?
- Q3) I think you might want to clarify early on that the usage of $f$-divergences here concerns the generalization of the regularization penalty term only, to distinguish it from other usages in LLM post-training where they have been used as the loss to minimize in a distribution matching objective (https://arxiv.org/abs/2302.08215). Also, not sure if this is relevant, but I also note that there was another contemporary paper that studied the $f$-divergence generalization of the penalty term in the context of RLVR: https://arxiv.org/abs/2509.07430.

---

> ### Author Response · Authors · 2025-11-20
>
> We thank the reviewer for their positive and thorough review.
>
>
> > Q1) Your "displacement-resistant" condition is a necessary, but is it sufficient to avoid likelihood displacement? If not, isn't the name "displacement-resistant" a bit misleading?
>
> The condition is not sufficient to avoid likelihood displacement. We chose the term “resistant” intentionally in the standard robustness sense used across ML and optimization: it indicates reduced susceptibility to a pathology, not absolute immunity. To avoid ambiguity, we have revised the Limitations paragraph to make it more explicit that finding a sufficient condition is left for future work.
>
>
> > Q2) Your displacement-resistant is quite simple. Wouldn't it predict that a simple change of variables, say t’=t-(1-e^-1)  for the reverse KL, work just as well?
>
> The function obtained by this additive change of variables for the reverse KL, i.e. $f(t)=(t-1+e^{-1}) \log (t-1+e^{-1})$, indeed attains a global minimum at $1$, and therefore satisfies our displacement-resistant condition. However, this function is not defined for $t < 1 - e^{-1}$ (since $\log$ is not defined for negative inputs) and therefore cannot be used as an $f$ for $f$-divergence (only functions that are defined on $[0,\infty)$ can be used).
>
> Therefore, our proposed $f(t)=(\log t)^2$ can be seen as the ”most simple” way to turn reverse KL into a displacement-resistant divergence in a way that actually yields a novel, valid loss.
>
>
> > Q3) I think you might want to clarify early on that the usage of $f$-divergences here concerns the generalization of the regularization penalty term only, to distinguish it from other usages in LLM post-training where they have been used as the loss to minimize in a distribution matching objective (https://arxiv.org/abs/2302.08215). Also, not sure if this is relevant, but I also note that there was another contemporary paper that studied the $f$-divergence generalization of the penalty term in the context of RLVR: https://arxiv.org/abs/2509.07430.
>
> Thank you for helping improve our work. We have added a clarification in the introduction that we use $f$-divergences as an alternative to the KL regularization term, rather than in a distribution-matching framework. We have also added both papers you mentioned to the related work section.
>
> > W2) The usage of a non-convex $f$  felt quite arbitrary, and there is no comparison with a convex displacement-resistant function. In a sense, the generalization to non-convex  $f$ might be interesting, but it's unclear from the content of this paper what can we gain from it.
>
> Our intention is not to argue that non-convex $f$ are universally better. Rather, our analysis identifies a class of $f$ that works perfectly fine under DPO-style training, and that convex functions are only a specific (but important) instance of this class. SquaredPO is merely the simplest instantiation we could find to empirically probe this class, showing that members of this class can indeed be used in $f$-DPO, and that they are generally as performant as the members of the previous most general class identified by the $f$-DPO paper (https://arxiv.org/abs/2309.16240).
>
>
> > W3) More in general, the comparison of the proposed technique is very limited as it only considers one baseline (DPO) and one dataset (TL;DR)...
>
> We agree that a broad empirical coverage is important. To support generality beyond the main TL;DR + Llama setup, Appendix C (included in the original submission) reports an additional setting with a different preference dataset (UltraFeedback) and a different reference model family (Qwen). The qualitative displacement trends and the relative behavior of SquaredPO vs. DPO remain consistent in this setting, suggesting the phenomenon and our mitigation are not specific to a single dataset or model.
>
> In addition, there are some additional experimental results we plan to share in the coming days following your remark (which are not ready yet for computational reasons).

---

> > ### Comment · Reviewer_Y1Xo · 2025-11-21
> >
> > I thank the authors for the clarifications and look forward to the additional experimental results.

---

> > > ### Author Response · Authors · 2025-11-25
> > >
> > > We thank the reviewer for their patience and kindly refer them to the official comment we have just posted to all reviewers with some additional empirical results.

---

### Official Review · Reviewer_iniP · 2025-10-29

**Soundness:** 3
**Presentation:** 3
**Contribution:** 3
**Rating:** 4
**Confidence:** 3

**Summary:**

Likelihood displacement has been know as one of the problems of DPO. This paper tries to solve this likelihood displacement issue by changing the KL divergenge regularization term in RLHF objective to nonconvex f-divergences. It theoretically shows that a convexity is not a necessary condition. From this theoretical finding, this paper suggests square of log function. It is both displacement resistant and DPO inducing. Empirical results on TL;DR and MT-Bench show that SQUAREDPO alleviates displacement while maintaining alignment performance comparable to standard DPO.

**Strengths:**

- Rigorous theoretical analysis.
- Easy to adapt loss (only regularization term is changed yet the validity is proved)

**Weaknesses:**

- Experiment is narrow.(model is only llama, the only baseline is naiveDPO)
- In experiment, the performance gain is marginal or even similar to naive DPO. Specifically, if the same performance for squareDPO be achieved with epoch 4 as DPO with epoch 1(Table1), why we should use SquareDPO?
- No performance analysis without LORA

**Questions:**

- How much displacement is mitigated?
How it improved the model performance empirically?

---

> ### Author Response · Authors · 2025-11-20
>
> We thank the reviewer for their comments and for their appreciation of our theoretical analysis.
>
> > In experiment, the performance gain is marginal or even similar to naive DPO. Specifically, if the same performance for squareDPO be achieved with epoch 4 as DPO with epoch 1(Table1), why we should use SquareDPO?
>
> While it is true that the performance of SquaredPO at epoch 4 is the same as DPO at epoch 1, it is important to highlight that this performance is also the same as SquaredPO at epoch 1 (where “same” here means that there might be differences in performance, but they are smaller than the error bar). In essence, what Table 1 shows us is:
>
> performance of SquaredPO after 1 epoch $\approx$ SquaredPO after 2 epoch $\approx$ SquaredPO after 4 epochs $>$ DPO after 2 epochs $>$ DPO after 4 epochs.
>
> This demonstrates that while both methods achieve comparable performance after one epoch, DPO rapidly degrades when trained longer, whereas SquaredPO’s performance remains stable.
>
> From a practical perspective, the number of training epochs is a hyperparameter whose optimal value may not be known in advance. Therefore, a method that is robust to overoptimization is of practical interest, as it reduces the need for techniques such as early stopping or extensive tuning of the number of epochs.
>
> We hope this answers the reviewer’s concerns, and we would be happy to answer any follow-up questions.
>
>
> > How much displacement is mitigated? How it improved the model performance empirically?
>
> We present a quantitative analysis of the mitigation of probability displacement by our method in the last four paragraphs of Section 5 and in Appendix C2. You can find there information about the mean and median winner probability throughout epochs, histograms of the whole training set, and statistics pertaining to a monotonicity phenomenon we identify. For example, in Section 5 we report that with DPO, $99.63\\%$ of the chosen responses whose probability decreased in the first epoch, continued decreasing monotonically in the three subsequent epochs, whereas with our method this proportion is $4.21\\%$ .
>
> Regarding how mitigation improved performance, empirically, our method matches DPO’s performance after one epoch and remains stable under continued training, whereas DPO degrades significantly (see Table 1). Thus, the performance benefit is robustness to over-optimization.
>
> Please feel free to let us know if there is anything specific you would like to know regarding the extent of mitigation beyond what we have reported.
>
>
> > Experiment is narrow.(model is only llama, the only baseline is naiveDPO)... No performance analysis without LORA
>
> We appreciate your concern regarding our experimental scope. We note that Appendix C includes an additional model/dataset setting beyond llama and TL;DR, namely, we report there experiments with Qwen and UltraFeedback. There are some additional experimental results we plan to share in the coming days following your remark (which are not ready yet for computational reasons).

---

### Official Review · Reviewer_YofB · 2025-10-31

**Soundness:** 3
**Presentation:** 4
**Contribution:** 2
**Rating:** 4
**Confidence:** 4

**Summary:**

The paper proposes exploring DPO variants that are based on a generalization of f-divergences that does not require convexity of f. They also analyze how these variants such as SquaredPO prevent probability displacement. They derive a set of functions that are DPO-inducing and a set of functions that are displacement resistant. They show that SquaredPO in particular is more robust to over-optimization and mitigates displacement while maintaining comparable performance to DPO.

**Strengths:**

Analysis - The paper provides a thorough analysis of the properties of different objectives and characterizes a wide range of possible objective functions. They also provide direct empirical comparisons of changes in likelihood as well as win rate and benchmark performance.

Clarity - The paper provides a clear presentation of ideas with detailed explanations. The theoretical definitions and interpretations walk through the key ideas and the experimental setup is well described and provides support for the claims.

**Weaknesses:**

Contributions - While the analysis is thorough and claims made are well supported, there is a lack of comparison to other methods that aim to achieve the same goal and it is unclear whether the convexity constraint is an issue. Figure 1 shows that there are multiple convex functions which are DPO-inducing and displacement-resistant, many of which have already been explored and have successfully mitigated over-optimization of displacement. As a result, without further comparison to these existing methods or a strong justification as to why the convexity constraint is limiting, it is unclear whether the paper provides a method or insights that goes beyond existing methods. The key ideas involved are also primarily extensions of existing ideas in f-DPO and the cited paper for Lemma 2.

**Questions:**

- Could you provide justification as to why the convexity of f may be a concern?
- Could you provide comparisons to existing methods such as f-DPO or Chi-PO?

---

> ### Author Response · Authors · 2025-11-20
>
> Thank you for your review and for acknowledging our work. We hope that our response below helps in addressing your questions and comments.
>
> > Without further comparison to these existing methods or a strong justification as to why the convexity constraint is limiting, it is unclear whether the paper provides a method or insights that goes beyond existing methods….
> Could you provide justification as to why the convexity of f may be a concern?
>
> It is not that we are claiming that the convexity of $f$ is a concern, but that by limiting our choices to convex generating functions $f$, we are excluding perfectly fine DPO-style algorithms, such as our SquaredPO. In other words, the $f$-DPO paper (https://arxiv.org/abs/2309.16240) did not generalize DPO as much as they could have, and we are fixing it, provably so, since we are providing the full characterization of which functions $f$ can be used and which cannot (Corollary 1).
>
>
>
>
> > Could you provide comparisons to existing methods such as $f$-DPO or Chi-PO?
>
> We have added a new appendix (page 14 in the revised version) with a detailed comparison with $\chi$PO. We thank the reviewer for this suggestion, which helped us enrich our work. Regarding $f$-DPO, we note that $f$-DPO is a general framework, and that $\chi$PO, vanilla DPO, and our SquaredPO are all instantiations of $f$-DPO (as we detail in the new Appendix).

---

> > ### Comment · Reviewer_YofB · 2025-11-21
> >
> > Thank you for the responses. I would like to provide a couple clarifications about my concerns.
> >
> > > Concern about convexity
> >
> > I appreciate the response explaining the goal of generalizing f-DPO. My question about justifications for exploring beyond convexity is due to some confusion as to why this generalization is important and whether non-convexity provides benefits that convex functions cannot. I believe this justification is necessary as utilizing non-convex functions can also introduce complications in optimization while existing methods as shown in Figure 1 already address displacement. Further comments on this aspect would be helpful.
> >
> > > Comparisons to existing methods
> >
> > I appreciate the comparisons of the methods in terms of the f-divergence landscape, but I would be more interested in an empirical comparison of squaredPO to existing displacement-resistant methods based on win-rate, likelihood displacement, or benchmark performance to ensure the theory aligns with practice. Furthermore, the paper would be strengthened if it could be more strongly shown that addressing likelihood displacement improves performance.
> >
> > While the benefit of SquaredPO over DPO is clear, I think whether the theoretical results/insights apply to comparisons between displacement-resistant methods is unclear. These evaluations would also justify some of the assumptions made. For example, Lemma 2 assumes that all responses in the dataset will have rewards upper bounded by the max out-of-sample reward or in other words, assumes that the trained policy must be in some constrained region. The actual training dynamics may not result in this assumption holding and more comprehensively showing that the degree of likelihood displacement does follow the trend seen in Figure 4/Lemma 2 would strengthen the paper.

---

> > > ### Author Response · Authors · 2025-11-25
> > >
> > > Thank you for clarifying your concerns.
> > >
> > > > concern about convexity
> > >
> > > We provide some additional notes on convexity in the context of $f$-DPO, hopefully addressing the concern that non-convex functions can introduce complications in optimization:
> > > 1. Perhaps counterintuitively, the resulting $f$-DPO loss (Equation 6 in our paper) might not be convex in the probabilities output by the model even when $f$ is convex. For example, both the original DPO loss and $f$-DPO with reverse KL, i.e., with $f(t)=-\\ln(t)$, are not convex in $\\pi_\\theta (y_l \\mid x)$, i.e., the probability the model gives to the loser response $y_l$.
> > > 2. Due to the non-convex architectures of modern LLMs, the probabilities output by the model, $\\pi_\theta (y_l \\mid x)$ and $\\pi_\\theta (y_w \\mid x)$, are not convex in the model’s learned parameters in the first place. Therefore, the $f$-DPO loss is not convex in the LLM’s parameters regardless of its convexity in $\\pi_\\theta (y_l \\mid x)$ and $\\pi_\\theta (y_w \\mid x)$.
> > >
> > > We argue that the above two remarks imply that aligning an LLM using $f$-DPO with a non-convex $f$ should not introduce more optimization complications than when using $f$-DPO with a convex $f$. Empirical evidence to this claim and to the doability of optimizing non-convex alignment losses includes the successful optimization of the SquaredPO loss reported in our paper, and the LLM-discovered DiscoPOP loss (https://arxiv.org/abs/2406.08414), which was also high-performing (and is not an instantiation of $f$-DPO).
> > >
> > >
> > >
> > > > Comparisons to existing methods
> > >
> > > Regarding showing that addressing likelihood displacement improves performance, we agree that this is relevant to our work. Therefore, we have added a paragraph to the related work section (first paragraph of this section in the new revision) to more clearly cite some literature that addresses the connection between displacement and performance. We thank the reviewer for this suggestion.
> > >
> > > Regarding comparisons to existing methods, we have performed an empirical comparison to $\\chi$PO, which we provide both in an official comment to all reviewers and in the newly uploaded revision of the paper.

---

### Official Review · Reviewer_Zbuk · 2025-11-01

**Soundness:** 3
**Presentation:** 3
**Contribution:** 3
**Rating:** 8
**Confidence:** 2

**Summary:**

This work dives deeper into DPO with f-divergences ($f$-DPO): (1) Proves more relaxed DPO-inducing condition, i.e., a sufficient condition that yields the $f$-DPO loss; (2) Proves a displacement-resistant condition, i.e., a necessary condition for $f$-DPO to avoid probability displacement issue; (3) proposes SQUAREDPO, a special case of $f$-DPO that satisfies the DPO-inducing and displacement-resistant conditions, and is empirically demonstrated to have better performance and mitigated displacement issue compared with vanilla DPO.

**Strengths:**

The perspective of looking at $f$-DPO is novel, with non-trivial theoretical results, which also yields a new DPO variant that is theoretically and empirically better than vanilla DPO. Many benchmarks are used in the experiments. There seem to be sufficient details to reproduce the experiments.

**Weaknesses:**

Figure 4 does not show significant performance difference between DPO and the proposed SQUAREDPO.

The paper could be better if

(1) Sufficient conditions for displacement-resistancy could be provided.

(2) $f$-DPO with more choices of $f$ could be empirically compared, such as $\chi^2$.

**Questions:**

(1) What is $\mathbb{R} _ {++}$ in Corollary 1?

(2) Both $\ln$ and $\log$ are used in the paper. Do they mean the same?

(3) Is there any sufficient condition to prevent probability displacement?

(4) What metrics are used in Figure 4?

---

> ### Author Response · Authors · 2025-11-20
>
> We sincerely thank the reviewer for the detailed and positive feedback. We now address the questions raised by the reviewer:
>
>  > (1) What is $\mathbb{R}\_{++}$ in Corollary 1?
>
> Thank you for the comment. In Corollary 1, $\mathbb{R}\_{++}$
>  denotes the set of strictly positive real numbers (the distinction between $\mathbb{R}\_+=[0,\infty)$ and $\mathbb{R}\_{++}=(0,\infty)$ is needed because, for example, the $f$ of the KL divergence, $f(t)=t \log t$, is continuous in $\mathbb{R}\_+$ but only differentiable in $\mathbb{R}\_{++}$). We have added a footnote in the paper explicitly defining this notation.
>
> > (2) Both $\ln$ and $\log$ are used in the paper. Do they mean the same?
>
> Yes, both denote the natural logarithm. We have standardized to $\log$ throughout and stated in Preliminaries that $\log$ is base $e$.
>
>
> > (3) Is there any sufficient condition to prevent probability displacement?
>
> We did not find a sufficient condition to prevent probability displacement. Establishing one remains an open problem, and we revised the Limitations paragraph to say so explicitly and leave it for future work.
>
> >(4) What metrics are used in Figure 4?
>
> Thank you for pointing this out. Figure 4 (now Figure 5 in the updated draft) reports scores on a 0–10 scale (higher is better) obtained using LLM-as-a-judge evaluation following the MT-Bench protocol, with OpenAI’s gpt-4o serving as the judge model. Scores are averaged within each MT-Bench category. We have updated the caption of this figure to make this information explicit.
>
>
> >The paper could be better if $f$-DPO with more choices of $f$ could be empirically compared, such as chi squared.
>
> We appreciate your concern regarding our experimental scope. There are some additional experimental results we plan to share in the coming days following your remark (which are not ready yet for computational reasons).

---

> > ### Comment · Reviewer_Zbuk · 2025-11-27
> > **Well solved my questions. I keep rating=8**
> >
> > Well solved my questions.
> > I keep rating=8.
> > Reviewer Zbuk

---

### Author Response · Authors · 2025-11-25

We thank all the reviewers for the ongoing discussion. A concern shared by multiple reviewers was that our original manuscript included a performance comparison with only one baseline. In particular, the reviewers asked about $\chi$PO, which is indeed highly relevant to our work. We now report additional experimental results showing that SquaredPO performs favorably and is more robust to over-optimization compared to both DPO and $\chi$PO.

These results have been incorporated into the newly uploaded revision (Section 5), and we also present them here for the reviewers’ convenience.

> Win Rate on TL;DR's validation set against the base model Meta-Llama-3-8B-Instruct when trained on TL;DR

| Epoch | SquaredPO winrate (%) | ChiPO winrate (%) | DPO winrate (%) |
|:-----:|:----------------------:|:------------------:|:----------------:|
|   1   |      50.8 ± 0.7        |     51.2 ± 0.8     |    51.8 ± 1.0    |
|   2   |      50.6 ± 1.1        |     48.9 ± 1.1     |    45.0 ± 1.1    |
|   4   |      51.0 ± 0.7        |     48.3 ± 1.1     |    34.7 ± 1.3    |


> Benchmark results (after 1 epoch on Meta-Llama-3-8B-Instruct with TL;DR)

| Method     | AlpacaEval 2 LC (%)        | AlpacaEval 2 WR (%)        | MT-Bench Score (1–10) |
|:----------:|:-------------:|:-------------:|:----------------------:|
| SquaredPO  | 29.2 ± 0.4    | 24.5 ± 0.3    | 7.924                  |
| χPO        | 29.3 ± 0.6    | 24.3 ± 0.4    | 7.900                  |
| DPO        | 29.6 ± 0.4    | 24.8 ± 0.4    | 7.925                  |


*(In LC, WR, and MT-Bench score, higher is better.)*

---

### Meta-Review · Area_Chair_8vQ2 · 2026-01-11

**Summary:**

This paper studies direct preference optimization (DPO)–style alignment methods for RLHF and revisits the conditions under which the underlying optimization problem remains tractable. The authors show that the convexity of the divergence used in the KL-regularized objective is not necessary, and instead introduce a broader DPO-inducing condition that characterizes tractability. They further identify a separate displacement-resistance condition that prevents probability collapse for preferred and dispreferred responses. Based on these insights, the paper proposes SquaredPO, a new loss function that satisfies both properties, and claims to offer stronger theoretical guarantees while achieving competitive empirical performance relative to standard DPO.

**Reviewer Concerns:**

- Concern 1: Unclear necessity of relaxing the convexity constraint
Provide full characterization of usable functions (Corollary 1). Additionally, explained that f-DPO loss is often non-convex in probabilities even with convex f, and LLM architectures are inherently non-convex, so non-convex f doesn't introduce additional optimization complications. Well addressed with both theoretical justification and practical optimization evidence.

- Concern 2: Missing comparisons to existing displacement-resistant methods (χ²-PO, f-DPO variants)
Authors added a detailed comparison with χ²-PO in the new appendix (page 14) and clarified that χ²-PO, vanilla DPO, and SquaredPO are all f-DPO instantiations. However, empirical comparison beyond performance metrics is initially limited. The authors promised additional experimental results. Partially addressed - some theoretical comparison added later.

- Concern 3: Limited experimental scope (single model/dataset, only DPO baseline)
Appendix C includes additional experiments (Qwen + UltraFeedback beyond main Llama + TL;DR). Added MT-Bench results and promised additional experiments. Later added χ²-PO empirical comparison and GPT-4o-mini experiments. Adequately addressed through appendix content and promised additions, though breadth is still limited compared to typical ICLR empirical papers.

- Concern 4: Marginal performance gains - unclear practical benefit
Authors clarified the key benefit is robustness to over-optimization: SquaredPO maintains stable performance across epochs 1-4 while DPO degrades significantly after epoch 1. Argued that this reduces the need for early stopping and hyperparameter tuning. Quantified displacement mitigation: 84% of DPO responses with decreasing probability continue monotonically decreasing vs. 49% for SquaredPO. This is sufficiently well addressed - reframed contribution from raw performance to stability/robustness.

- Concern 5: Displacement-resistant condition is only necessary, not sufficient
The authors acknowledged that the condition is necessary but not sufficient, intentionally using "resistant" (not "immune") in the standard robustness sense. The authors later revised the Limitations section to clarify that finding a sufficient condition is future work. Honestly addressed - limitation acknowledged clearly.


Recommendation: WEAK ACCEPT (borderline)
Reasoning:
Strengths:

Based on the theoretical contribution, i.e., full characterization of DPO-inducing functions beyond the convex class (Corollary 1), is genuinely novel and rigorous. The practical relevance and robustness to over-optimization are real practical concerns. Authors acknowledge necessary vs. sufficient conditions and respond constructively to concerns.

On the other hand, even with additions, experiments remain narrow (2-3 model families, 2-3 datasets) for an ICLR empirical/applied paper
Incremental practical impact: SquaredPO matches but doesn't exceed DPO performance; the benefit is stability rather than better outcomes. Also, there is no direct evidence that displacement reduction causes performance benefits, and experiments are missing key comparisons: Limited head-to-head with other displacement-resistant methods beyond χ²-PO

Overall, the work advances understanding of DPO variants in a principled way. The paper would definitely benefit from expanded experiments. Please incorporate all improvements and clarifications in the final version.

**Reviewer Scores:**

I think the rebuttal sufficiently addressed the concerns raised by iniP - HeeSun Bae, and YofB - Shawn Im (both with score 4), which should approve an acceptance.

---

### Decision · Program_Chairs · 2026-01-26

Accept (Poster)